# Evaluation of Cytotoxicity and Metabolic Profiling of *Synechocystis* sp. Extract Encapsulated in Nano-Liposomes and Nano-Niosomes Using LC-MS, Complemented by Molecular Docking Studies

**DOI:** 10.3390/biology13080581

**Published:** 2024-07-31

**Authors:** Lamya Azmy, Ibraheem B. M. Ibraheem, Sulaiman A. Alsalamah, Mohammed Ibrahim Alghonaim, Ahmed Zayed, Rehab H. Abd El-Aleam, Soad A. Mohamad, Usama Ramadan Abdelmohsen, Khaled N. M. Elsayed

**Affiliations:** 1Botany and Microbiology Department, Faculty of Science, Beni-Suef University, Beni-Suef 62511, Egypt; lamyaazmy@gmail.com (L.A.); ibraheemborie@science.bsu.edu.eg (I.B.M.I.); 2Department of Biology, College of Science, Imam Mohammad Ibn Saud Islamic University, Riyadh 11623, Saudi Arabia; saalsalamah@imamu.edu.sa (S.A.A.); mialghonaim@imamu.edu.sa (M.I.A.); 3Pharmacognosy Department, Faculty of Pharmacy, Tanta University, Tanta 31527, Egypt; ahmed.zayed1@pharm.tanta.edu.eg; 4Pharmaceutical Chemistry Department, Faculty of Pharmacy, Modern University for Technology and Information MTI, Cairo 11571, Egypt; rehab.hamed@pharm.mti.edu.eg; 5Clinical Pharmacy Department, Faculty of Pharmacy, Deraya University, New Minia 61111, Egypt; soad.ali@deraya.edu.eg; 6Deraya Center for Scientific Research, Deraya University, New Minia 61111, Egypt; usama.ramadan@mu.edu.eg; 7Pharmacognosy Department, Faculty of Pharmacy, Minia University, Minia 61519, Egypt

**Keywords:** metabolic profiling, cyanobacteria, cytotoxicity, liposomes, nanotechnology, niosomes, *Synechocystis* sp.

## Abstract

**Simple Summary:**

Cancer is the second leading cause of death worldwide, warranting the development of safer, more effective therapies. Liposomes and niosomes can be considered excellent drug delivery systems due to their ability to load all compounds and reduce the toxicity of the loaded drug without reducing its effectiveness. *Synechocystis* sp. is a unicellular, freshwater blue-green algae strain that has many bioactive compounds that qualify for use in industrial, pharmaceutical, and other fields. This study used nano-liposomes and nano-niosomes to deliver *Synechocystis* sp. extract against human colon, ovarian, and breast cancer cell lines. The results demonstrated potential activities against human colon, ovarian, and breast cancer cell lines. A total of 22 compounds were identified through the metabolic profiling of an extract, as it is a tool aimed at identifying biologically active compounds in the sample. Subsequently, the molecular docking of these compounds was studied, which is the most effective way to study the compound’s effectiveness on a specific disease by determining the binding affinity. The results showed that compounds 21, 6, 7, 8, 12, and 19 have a high degree of correlation. Finally, these results represent a promising step toward developing effective cancer treatments. Future research should translate this approach into successful anticancer drugs through participation in clinical trials.

**Abstract:**

Liposomes and niosomes can be considered excellent drug delivery systems due to their ability to load all compounds, whether hydrophobic or hydrophilic. In addition, they can reduce the toxicity of the loaded drug without reducing its effectiveness. *Synechocystis* sp. is a unicellular, freshwater cyanobacteria strain that contains many bioactive compounds that qualify its use in industrial, pharmaceutical, and many other fields. This study investigated the potential of nano-liposomes (L) and nano-niosomes (N) for delivering *Synechocystis* sp. extract against cancer cell lines. Four different types of nanoparticles were prepared using a dry powder formulation and ethanol extract of *Synechocystis* sp. in both nanovesicles (N1 and N2, respectively) and liposomes (L1 and L2, respectively). Analysis of the formed vesicles using zeta analysis, SEM morphological analysis, and visual examination confirmed their stability and efficiency. L1 and L2 in this investigation had effective diameters of 419 and 847 nm, respectively, with PDI values of 0.24 and 0.27. Furthermore, the zeta potentials were found to range from −31.6 mV to −43.7 mV. Regarding N1 and N2, their effective diameters were 541 nm and 1051 nm, respectively, with PDI values of 0.31 and 0.35, and zeta potentials reported from −31.6 mV to −22.2 mV, respectively. Metabolic profiling tentatively identified 22 metabolites (1–22) from the ethanolic extract. Its effect against representative human cancers was studied in vitro, specifically against colon (Caco2), ovarian (OVCAR4), and breast (MCF7) cancer cell lines. The results showed the potential activities of the prepared N1, N2, L1, and L2 against the three cell lines, where L1 had cytotoxicity IC50 values of 19.56, 33.52, and 9.24 µg/mL compared to 26.27, 56.23, and 19.61 µg/mL for L2 against Caco2, OVCAR4, and MCF7, respectively. On the other hand, N1 exhibited IC50 values of 9.09, 11.42, and 2.38 µg/mL, while N2 showed values of 15.57, 18.17, and 35.31 µg/mL against Caco2, OVCAR4, and MCF7, respectively. Meanwhile, the formulations showed little effect on normal cell lines (FHC, OCE1, and MCF10a). All of the compounds were evaluated in silico against the epidermal growth factor receptor tyrosine kinase (EGFR). The molecular docking results showed that compound 21 (1-hexadecanoyl-2-(9Z-hexadecenoyl)-3-(6′-sulfo-alpha-D-quinovosyl)-sn-glycerol), followed by compounds 6 (Sulfoquinovosyl monoacylgycerol), 7 (3-Hydroxymyristic acid), 8 (Glycolipid PF2), 12 (Palmitoleic acid), and 19 (Glyceryl monostearate), showed the highest binding affinities. These compounds formed good hydrogen bond interactions with the key amino acid Lys721 as the co-crystallized ligand. These results suggest that nano-liposomes and nano-niosomes loaded with *Synechocystis* sp. extract hold promise for future cancer treatment development. Further research should focus on clinical trials, stability assessments, and pharmacological profiles to translate this approach into effective anticancer drugs.

## 1. Introduction

Cyanobacteria, also called “blue-green algae”, are photosynthetic or photoautotrophic prokaryotes commonly found in a wide range of environments, including terrestrial and marine habitats [1]. They also have different morphologies, like unicellular and filamentous forms, where the unicellular morphologies include single, suspended or benthic, or aggregate cells. However, filamentous morphologies may be thin or thick, single trichomes, or bundles with or without a sheath [2,3,4]. Cyanobacteria can grow independently or as symbionts with other organisms. In addition, there are many distinct cyanobacteria genera, each with unique properties. Common cyanobacteria genera that have been recorded in freshwater environments include *Synechococcus*, *Anabaena*, *Rivularia*, *Gloeotrichia*, *Oscillatoria*, *Cylindrospermopsis*, *Ahanizomenon*, *Planktothrix*, *Scytonema*, *Tolypothrix*, *Merismopedia*, and *Microcystis* [5]. Among the diverse cyanobacterial strains, *Synechocystis* sp., a photoautotrophic cyanobacterial strain mostly found in aquatic environments, is one of the most extensively studied model organisms [6].

In addition to their photosynthetic abilities, cyanobacteria contain numerous bioactive molecules as pigments, like phycobiliproteins and mycosporine-like amino acids (MAAs), which are important molecules due to their antibacterial and antioxidant activities. They also include carotenoids like β-carotene, fucoxanthin, and zeaxanthin, which have anticancer effects against melanoma cell lines. Furthermore, cyanobacteria contain polysaccharides, which have anti-inflammatory properties [5]. Mostly, cyanobacterial bioactive molecules from different families are involved in a wide spectrum of different fields, including cosmetology, agriculture, and pharmaceuticals (e.g., for their immunosuppressant, anticancer, antibacterial, antiprotozoal, antifungal, anti-inflammatory, antimalarial, antitumor, and antiviral activities) [7]. *Synechocystis* sp. possesses unique properties, as it contains various potential bioactive compounds as antioxidant, anticancer, and anti-inflammatory agents, such as phenols, alkaloids, fatty acids, terpenoids, and carotenoids [3,8,9], with an emphasis placed on its anticancer properties and its potential as a powerful anticancer agent [10,11]. Its fatty acids have been shown to include monogalactosyldiacylglycerol, which has promising activity against breast cancer cells [12].

Therefore, *Synechocystis* sp., with its rich collection of biologically active compounds and its high growth rate, which helps to produce biomass easily, is considered a unique and promising opportunity in nanotechnology. It can be considered a sustainable source of bioactive compounds due to its rapid growth and ease of cultivation [13,14,15]. Therefore, *Synechocystis* sp. has been harnessed in nanotechnology, leading to the successful production of various nanoparticles which are effective in antimicrobial and anticancer applications [16].

Nanoparticles (NPs) are a great field of science that can be used in many other fields. A nanoparticle can be defined as a very small molecule that has unique properties. Because of its small size, it exhibits unique properties, such as being a drug conductor in addiction. Its special physical, chemical, and biological properties allow it to cross cells and tissues and reach the target location to influence it [17]. NPs have achieved great success in medicine and in drug delivery, improving drug efficiency and treating a wide range of diseases by overcoming the limitations of traditional treatments, including cancer. They enable the targeting of specific targets, such as particular organs or tissues. The use of nano-encapsulation technology also allows for the use of natural substances that are effective against cancer and which enhance their bioavailability [18,19].

NPs exhibit various shapes, such as spherical, spiral, cylindrical, or even irregular shapes. In addition, they can be classified into the following three basic groups: organic, inorganic, and carbon-based [20]. Carbon-based NPs include molecules made of carbon only, while inorganic NPs are manufactured by extracting some metals or alloys. They can all be used in various fields, from drug delivery to biological applications, including various medical fields [20,21,22,23,24]. As for organic NPs, they contain compounds made from proteins, fats, carbohydrates, or any other organic materials. What distinguishes this type is that they are mostly non-toxic, since they are dissolved in the body and are easy to get rid of, which endows them with great importance in the pharmaceutical field. Examples of this kind of nanoparticle include liposomes and niosomes. Generally, NPs have achieved great success in the field of medicine and drug delivery, improving the efficiency of medicines, treating a wider range of diseases, and producing various nutritional materials [20,25,26].

Liposomes consist of spherical or multilayered spherical vesicles which are formed by diacyl-chain phospholipids (lipid bilayer) in aqueous solutions. The membrane is a bilayer phospholipid membrane that has a hydrophobic tail and a hydrophilic head because amphiphilic structures are formed [27]. Also, liposomes are often used in pharmaceutical research as drug delivery systems, and have unique prorates, being more powerful for drug delivery systems in view of their structural versatility as well as their biocompatibility and biodegradability, in addition to their non-toxic and non-immunogenicity nature [28]. Niosomes are nonionic surfactant vesicles that can have unilamellar, multilamellar, or oligolamellar vesicular structures. Its components include cholesterol and nonionic surfactants, while its vesicle is made up of a hydrophobic tail and a hydrophilic head, which can be used for various medical applications [29,30,31]. Niosomes and liposomes show highly similar structures and effectiveness in drug delivery. However, niosomes are superior, as they are less expensive, more stable, and easy to preserve and store [27,30].

The efficacy of nano-liposome and nano-niosome technology in cancer treatment and drug delivery is widely acknowledged, making it a promising avenue for research [32,33,34,35]. Researchers have previously updated nano-liposomes loaded with doxorubicin and with folic acid, which resulted in additional anticancer activity [33]. Moreover, curcumin-loaded liposomes also showed superiority against breast cancer [36]. Nanosomal particles also showed the potential to fight breast cancer when they were loaded with curcumin anticancer drugs [37,38].

Some recent studies have achieved promising success in their applications against cancer, specifically breast cancer. Nanosomal particles are considered a smart device for delivering anticancer drugs, as they have many advantages, such as increasing the effectiveness of anticancer compounds, reducing their toxicity, the ability to release drugs for a long time, ease of preparation, high biological compatibility, and others. Their potential can also be enhanced by discovering new formulations from safe sources. Therefore, this field needs further study and exploration [39,40,41]. Notably, the application of *Synechocystis* sp. with nano-liposomes or nano-niosomes remains unexplored, despite previous studies employing liposomes with various types of blue-green algae. For instance, liposomes have been successfully manufactured using Spirulina platensis [42]. Its antioxidant and antibacterial activity was demonstrated to be significantly higher after encapsulation compared to the algae [43].

Cancer ranks second in terms of the number of deaths around the world, with an estimated 9.6 million deaths recorded in 2018 according to the World Health Organization [44]. Particularly, colon cancer is a life-threatening disease, especially for older people [45]. Colon cancer ranks third among cancer deaths in the Western Hemisphere [46]. Furthermore, the most prevalent cancer that claims the lives of many women is breast cancer [47]. Ovarian cancer ranks fifth in the number of deaths among women [48]. Thus, it is crucial to raise awareness of these problems, which endanger the lives of thousands of people worldwide, and work towards developing safer, more effective therapies.

The focus on epidermal growth factor receptor tyrosine kinase (EGFR) has been chosen based on prior literature suggesting its involvement in pathways relevant to the identified compounds [49,50,51,52]. The achieved cytotoxic activity of the crude extract needs to be explained on a molecular level. Therefore, the dereplicated compounds were in silico evaluated against the epidermal growth factor receptor tyrosine kinase (EGFR). EGFR, a receptor tyrosine kinase within the ErbB family, requires ligand binding to activate its tyrosine kinase domain, initiating signaling cascades crucial for cell proliferation, angiogenesis, migration, survival, and adhesion. Given the significance of these pathways in cancer cell survival, EGFR stands as a valuable target in treating many types of cancers, such as lung, breast, kidney, and colorectal carcinoma metastases [53,54].

In breast cancer, EGFR overexpression correlates with estrogen receptor loss and poor prognosis, suggesting its role in promoting growth in estrogen receptor-positive breast cancer cells, even in the absence of estrogen. Although MCF7 cells express lower levels of EGFR compared to other breast cancer cell lines, EGFR signaling pathways still significantly affect MCF7 cell biology [55,56,57].

In ovarian cancer, EGFR expression and activation correlate with tumor progression and poor prognosis. Studies indicate the potential efficacy of EGFR inhibitors such as cetuximab or gefitinib in treating ovarian cancer by disrupting EGFR-mediated signaling pathways. This suggests a plausible relationship between EGFR and OVCAR4 cells, implicating EGFR signaling in ovarian cancer progression and therapy resistance [58,59].

As previously mentioned cyanobacteria contain many bioactive compounds and can serve humanity in various ways. They can be used in potential drug formulations for treating cancer [7], and previous studies have focused on manufacturing inorganic NPs using blue-green algae [60,61,62]. Other studies have focused on the potential of *Synechocystis* sp. extract as a promising candidate against cancer [10], in addition to the effectiveness of nano-liposomal and nano-niosomal systems in delivering anticancer drugs [33,34]. Nano-liposomes and nano-niosomes can capsulate bioactive compounds from cyanobacteria sp. extracts, increasing their solubility, stability, and specific delivery to the tumor target. This makes the drug accumulate in a greater concentrations in the cancer cell, enhancing its cytotoxicity while preserving healthy cells [63,64]. Research gaps exist in studying the effect of nano-liposomes and nano-niosomes loaded with *Synechocystis* sp. on cancer cells. This is a unique opportunity for our study to contribute significantly to this field. This study investigates the anticancer potential of nano-liposomes and nano-niosomes loaded with *Synechocystis* sp. extract against Caco2 (colon), OVCAR4 (ovarian), and MCF7 (breast) cancer cell lines. We employed metabolomics and in silico analysis to identify active metabolites and their potential mechanisms of action.

## 2. Materials and Methods

### 2.1. Microalgae Cultures

Firstly, *Synechocystis* sp. (axenic cyanobacterial cultures) was grown on Wuxal (WM) media with tap water [60,65]. It is a universal liquid plant fertilizer that contains 8% N, 8% P_2_O_5_, 6% K_2_O, 0.01% B, 0.004% Cu, 0.02% Fe, 0.012% Mn, and 0.004% Zn (Wilhelm Haug GmbH and Co. KG, Germany). The ratio of the culture medium to tap water during the experiments was 1 mL Wuxal medium/1 L H_2_O. Then, *Synechocystis* sp. was cultured in 250 mL Erlenmeyer flasks to monitor its growth under standard conditions at 28 °C. The culture was exposed to light for 10 h and preserved in the dark for 14 h. Then, the subculture was left for two weeks, with daily monitoring and replacing with new media every 24 days. After that, the subculture was carefully transferred to complete the growth in a photobioreactor system to obtain a large amount of algal biomass. This is also called the ‘hanging bag’ system [66]. The subculture was left for one month, and daily monitoring was performed under standard conditions. Subsequently, the biomass was harvested, and then dried with air and incubated at room temperature for 48 h; the final product was 1.8 g [67].

### 2.2. Extract Preparation

Air-dried *Synechocystis* sp. powder was aseptically weighed and divided equally into two halves. One half (0.9 g) was transferred to a sterile flask containing 100 mL of ethanol (99%). The mixture was stirred on a magnetic stirrer at 200–300 rpm for 24 h at room temperature to facilitate the efficient extraction of the bioactive compounds. After stirring, it was filtered with Whatman No. 1 filter paper 3 times. The ethanol was evaporated in dry air for 48 h to yield a dried ethanol extract powder. The other half (0.9 g) of the *Synechocystis* sp. powder was left as non-extracted. They were kept in sterile containers to use them in the following steps [60].

### 2.3. Metabolic Profiling and Peak Annotation in LC-MS

The metabolomic analysis utilized a Synapt G2 HDMS quadrupole time-of-flight hybrid mass spectrometer (Waters, Milford, CT, USA) connected to an Acquity Ultra Performance Liquid Chromatography system. Chromatographic separation was carried out using a BEH C18 column (Waters, Milford, CT, USA) with dimensions of 2.1 mm in diameter, 10 cm in length, and a particle size of 1.7 μm, along with a guard column (2.1 cm × 5 mm, 1.7 μm). A gradient elution of an acetonitrile–water system, starting at 5% acetonitrile and ending at 100% acetonitrile, was performed at a flow rate of 0.3 mL/min. The column was washed with water containing 0.1% formic acid (*v*/*v*) and maintained at 40 °C. The raw data were processed into sliced positive and negative ionization files using mass spectrometry-converting software. These files were then submitted to the data chromatogram builder in the software MZmine 2.10 (Okinawa Institute of Science and Technology Graduate University, Japan) [68], which was used to identify the peaks in the samples and blanks. Mass ion peaks were separated using the following steps: Raw Data Methods → Peak detection → Mass detection. The MS level was adjusted, providing that the detector threshold was larger than the noise level by 1.0 × 10^4^, with the *m*/*z* tolerance at 0.001 *m*/*z* or 5.0 ppm and a minimum time span of 0.2 min. After that, chromatogram deconvolution was carried out to identify each individual peak. Additionally, a gap-filling peak finder was used to identify the absent peaks. Both a complicated search and an adduct search were used. Next, peak identification and chemical formula prediction were applied to the processed data set. The Dictionary of Natural Products (DNP) database was used to dereplicate the data sets for the *Synechocystis* sp. extract’s positive and negative ionization modes [69].

### 2.4. Nanoparticle Synthesis

#### 2.4.1. Nano-Liposome and Nano-Niosome Preparation

Equal amounts of pure powder and full extract were formulated in both niosome (N1 and N2, respectively) and liposome (L1 and L2, respectively) nanovesicles. Four different NPs were prepared and were characterized as follows:

In short, niosomal preparations were developed by individually combining complete powder as well as an extract in weighed amounts with lecithin (the lecithin was L-α-Lecithin, soybean, 1,2-Diacyl-sn-glycero-3-phosphocholine, 3-sn-Phosphatidylcholine, L-α-Lecithin, Azolectin, PC, L-α-Lecithin, and egg yolk, highly purified, purchased from Sigma Aldrich, St. Louis, MO, USA) and cholesterol in a glass vial (5 mL). The ratio of lecithin to cholesterol was 1:1. Then, 95% ethanol was added, the container was covered, and it was heated in a thermostatic water bath (60 ± 2 °C) until the surfactant mixture was completely dissolved. Phosphate-buffered saline (PBS) with a pH of 7.4 was added to this solution, and the combination was further heated in a thermostatic water bath until it became transparent. It was then subjected to a 45 s, 20 W sonication (Omni-Ruptor 4000, Omni International Inc., Kennesaw, GA, USA) with drug-loaded niosomes, which formed a precipitate, and was kept at a temperature of 4–8 °C for the investigations [70]. On the other hand, to prepare the liposomes using the film method [71,72], additional equal amounts of both the whole powder and extract were combined with soybean phospholipids/cholesterol (9:1) that had previously been dissolved in chloroform. After mixing the total solutions, the solvent was then evaporated under a vacuum using a rotary evaporator at 45 ± 0.5 °C for 30 min and dried; the product was then kept at room temperature for 24 h. After drying, the vesicles were produced and kept at a temperature of 4–8 °C for the investigations.

Further steps were taken to dissolve the film with phosphate-buffered saline (PBS) with a pH of 7.4 for characterization.

#### 2.4.2. Vesicle Characterization

##### Visual Observation and Physicochemical Evaluation

In transparent containers, all of the prepared vehicles were transferred to be examined visually for color, transparency, and turbidity, if found. Results for transparency and turbidity were analyzed based on the containers’ top and side views after the experiment was carried out in three duplicates. The pH was measured using an electronic pH meter (the LCD Benchtop Lab pH Meter Kit with Refillable pH Electrode).

##### Vesicle Size Measurement and Zeta Potential Measurement

The prepared particles were analyzed for their particle size and size distribution, in terms of the average volume diameters and polydispersity index, with photon correlation spectroscopy using a particle size analyzer and Dynamic Light Scattering (DLS) (Zetasizer Nano ZN, Malvern Panalytical Ltd., Malvern, UK) at a fixed angle of 173° at 25° C. Samples were analyzed in triplicate. The same equipment was used to determine the zeta potential. The working principle of the instrument is electrophoretic light scattering (ELS), which determines the electrophoretic movement of charged particles under an applied electric field from the Doppler shift of scattered light, which was used for the zeta potential determination.

##### SEM Microscopic Analysis

To examine the vesicles’ morphologies at higher magnification levels, scanning electron microscopy (SEM) was also used. SEM (JSM-IT 200, JEOL Ltd., Tokyo, Japan) made it possible to characterize the morphology of the vesical surface. Fixed gold sputtering was carried out after coating the liposomal samples with ION COAT JFC-1100, JEOL Ltd., Tokyo, Japan. Different magnifications were used during the analysis.

### 2.5. Cytotoxicity Assay

The cytotoxic activity of the four nano-formulations (L1, L2, N1, and N2) was evaluated. The three human cancer cell lines, human Caco-2, MCF-7, and OVCAR4, and the three normal cell lines, FHC, OCE1, and MCF10a, were obtained from The American Type Culture Collection (ATCC, Manassas, VA, USA). The cells were cultured using DMEM (Invitrogen/Life Technologies, Waltham, MA, USA) supplemented with 10% FBS (Hyclone), 10 µg/mL insulin (Sigma-Aldrich (Merck KGaA, Darmstadt, Germany)), and 1% penicillin–streptomycin. All of the other chemicals and reagents were from Sigma or Invitrogen. The cells were plated (cell density of 1.2–1.8 × 10,000 cells/well) in a 100 µL complete growth medium + 100 µL of the tested compound per well in a 96-well plate for 24 h before the MTT assay. The following five different concentrations of each nano-formulation were used: 100 μg, 25 μg, 6.3 μg, 1.6 μg, and 0.4 μg.

#### 2.5.1. Cell Culture Protocol

The culture medium was removed and collected in a centrifuge tube. The cell layer was then briefly rinsed with a 0.25% (*w*/*v*) trypsin solution containing 0.53 mM EDTA to eliminate any residual serum that might contain trypsin inhibitors. Next, 2.0 to 3.0 mL of fresh trypsin–EDTA solution was added to the culture vessel. The cells were observed under an inverted microscope until they detached from the surface (typically within 5–15 min). To inactivate the trypsin, 6.0 to 8.0 mL of complete growth medium was added, and the cell suspension was gently pipetted. The cell suspension was then combined with the initial culture medium collected in the centrifuge tube from step 1. This combined solution was centrifuged at approximately 125× *g* for 5–10 min. The supernatant was discarded, and the remaining cell pellet was resuspended in a fresh growth medium. Appropriate aliquots of the cell suspension were then added to new culture vessels for further incubation. The cultures were incubated for 24 h at 37 °C. Following incubation, the cells were treated with serial concentrations (100 μg, 25 μg, 6.3 μg, 1.6 μg, and 0.4 μg) of the test formulations and doxorubicin as a positive control, and then incubated for 48 h at 37 °C. The plates were subsequently examined under an inverted microscope, and the MTT test was performed [73].

#### 2.5.2. MTT–Cytotoxicity Assay Protocol

To achieve optimal results, cells in the log phase of growth were used, with a final density not exceeding 10^6^ cells/cm^2^. Each test incorporated a blank well containing a complete medium without cells. First, cultures were removed from the incubator and transferred to a laminar flow hood or another sterile work environment. Each vial of MTT [M-5655] was reconstituted with 3 mL of medium or a balanced salt solution lacking phenol red and serum. Subsequently, reconstituted MTT was added, equaling 10% of the culture medium volume. The cultures were then returned to the incubator for 2–4 h, with the specific duration depending on the cell type and the maximum cell density (an incubation period of 2 h was typically sufficient but could be extended for low cell densities or cells with lower metabolic activity). Consistent incubation times were crucial for accurate comparisons. Following the incubation period, cultures were removed from the incubator. The resulting formazan crystals were dissolved by adding an equal volume of MTT Solubilization Solution [M-8910] to the original culture medium volume. Gentle mixing on a gyratory shaker facilitated dissolution. In some cases, particularly with dense cultures, additional pipetting up and down (trituration) might be necessary to ensure complete dissolution of the MTT formazan crystals. Finally, the absorbance was measured spectrophotometrically at a wavelength of 570 nm. The background absorbance of the multiwell plates was measured at 690 nm and subtracted from the 570 nm measurement. Tests performed in multiwell plates could be read using a compatible plate reader, or the contents of individual wells could be transferred to cuvettes of appropriate size for spectrophotometric measurement; each well’s optical density (OD) was measured with an ELISA microplate reader. The following calculation was used: “% Cell Viability = [O.D. (treated cells)/O.D. (control cells)] × 100. The IC50 values were calculated using a linear equation, (X-axis: log conc, Y-axis: “% Cell Viability) (y = mx + b) (y = 0.5), using Excel’s regression analysis tools, where IC50 = (0.5 − b)/m [73,74].

#### 2.5.3. Statistical Analysis

The software Minitab 17 (Minitab Inc., College Station, PA, USA) was used to analyze the data obtained from the MTT assay. A one-way ANOVA was performed, and then Tukey’s Honestly Significant Difference (HSD) test was used for post hoc pairwise comparisons [75].

### 2.6. Molecular Docking

AutoDock Vina 1.1.2. software was employed for all molecular docking experiments (Seeliger and de Groot, 2010). The docking studies aimed to evaluate the binding affinity of isolated compounds against the epidermal growth factor receptor (EGFR). The EGFR crystal structure complexed with the 4-anilinoquinazoline inhibitor Erlotinib (PDB ID: 1M17) was obtained from the Protein Data Bank (https://www.rcsb.org/structure/1M17) (accessed on 22 July 2024). This specific crystal structure was selected due to its high-resolution data and the presence of a co-crystallized ligand, which facilitated the accurate determination of the binding site.

The binding site was defined based on the coordinates of the co-crystallized ligand within the enzyme’s active site. The docking grid box parameters for the EGFR were set as follows: x = 22.379, y = 20.9017, and z = 52.925, with a grid box size of 10 Å. This grid box size was chosen to ensure the entire binding pocket was encompassed within the docking simulations, allowing for a comprehensive exploration of potential binding interactions.

To ensure the reliability of the docking protocol, we performed a validation step by re-docking the co-crystallized ligand into the EGFR binding pocket. The resulting poses exhibited root mean square deviations (RMSDs) of 1.12 Å from the original crystallized ligand positions, indicating the accuracy of our docking setup. Achieving an RMSD close to 1 Å suggests that the docking protocol can reliably reproduce the experimentally observed binding mode, thereby enhancing confidence in the docking results for the isolated compounds.

Ten docking poses were generated for each compound, and the best-ranked pose based on the binding affinity score was selected for further analysis. The scoring function employed by AutoDock Vina evaluates the binding free energy, providing insights into the potential binding strength and stability of the ligand–receptor complex.

Docking poses were analyzed and visualized using Biovia Discovery Studio 21.1 software (Dassault Systèmes BIOVIA), which facilitated examining molecular interactions, including hydrogen bonds, hydrophobic interactions, and pi-pi stacking between the ligands and the active site residues of the EGFR.

Based on their structural diversity and potential bioactivity, the isolated compounds were selected for docking, as indicated by preliminary in vitro assays. These compounds represent a range of chemical scaffolds, allowing for a comprehensive evaluation of their interactions with the EGFR.

## 3. Results

### 3.1. LC-MS and Metabolite Identification

LC-MS has emerged as the most complete approach for measuring a wide variety of metabolites. The most widely utilized reversed-phase columns for LC gradient separation are C18 or C8. Additionally, the development of ultra-performance LC increased the technique’s resolution, sensitivity, and throughput [76].

The negative and positive MS-based metabolic profiling of the investigated *Synechocystis* sp. tentatively identified 22 metabolites (1–22) in accordance with the DNP and those previously reported in *Synechocystis* sp. (Table 1). The chemical structures are also illustrated in Figure 1. Moreover, the metabolite retention times are consistent with the use of reversed-phase UPLC, since the polar metabolites (e.g., sugars and phenolic compounds) appeared in the first half of the chromatogram, while the non-polar metabolites (e.g., depsipeptides and high-molecular-weight fatty acids) appeared later in the chromatogram in negative and positive modes, as demonstrated in Figure 2A,B, respectively.

### 3.2. Nanoparticle Vesicle Characterization

#### 3.2.1. Visual Examination

The niosomal and liposomal emulsions were evaluated for color, transparency, clarity, pH, and homogeneity. The pH was 7.0 ± 1.0, the color was from white to yellowish green, as shown in Figure 3, and clear without any flocculation. These results indicate that the formulated nanoparticles are physiochemically good. Physiochemically, the organoleptic properties have a clear appearance, no agglomerated or separate layers depending on sight, and are clear in the formulated dispersion. The stable physiological pH is considered close to a pH of 7.4 [110].

#### 3.2.2. Size and Zeta Analysis

The sizes of the liposomal and niosomal dispersions varied from nanometers to micrometers, with more than one peak that returned to the preparation process and given the nature of the extract, as shown in Figure 4. Liposomes L1 and L2 in this investigation had 419 and 847 nm effective diameters, respectively (Table 2). The liposomes’ PDI values were 0.24 and 0.27, respectively, demonstrating their polydispersity and high population homogeneity, as shown in Figure 4, demonstrating the size distribution of different vesicle dispersions versus the intensity. Furthermore, the zeta potential was highly stable, ranging from −31.6 mV to −43.7 mV. Regarding niosomes N1 and N2, their dispersion sizes ranged from nanometers to micrometers. According to Table 2 and Figure 5, the effective diameters were 541 nm and 1051 nm, respectively. The noisomes’ PDI values were 0.31 and 0.35, respectively, demonstrating their polydispersity and high population homogeneity. Conversely, the zeta potential was reported from −31.6 mV to −22.2. mV.

The zeta potential records the stability of the formulations and measures the difference in the charges in nanoparticle colloidal solutions to ensure the stability, which has a reference number of >±30 mV [40,111]. Additionally, the physical stability was confirmed, with the PDI measurement falling within the acceptable range, which provided reassurance about the formulations’ stability [112].

#### 3.2.3. SEM Morphological Analysis

The morphologies of the produced liposomes and niosomes were examined using SEM. Figure 6A,B indicate that the particle surfaces are rough and that the nano-liposomes have a spherical and regular structure without surface pores. Figure 6B demonstrates a bigger surface area with more fibers that connect back to the various extract components.

Figure 7A,B show the SEM images of the niosomes in aqueous solutions, and their morphologies are spherical and regular in shape, with clear membranes and backgrounds (Figure 7A), whereas the images in Figure 7B show more blurred outer membranes and a crowded background, which are reflected the extract’s nature.

### 3.3. Cytotoxicity Results

All of the results from the different cell lines are significantly different (*p* ≤ 0.05), as is the grouping information using the Tukey method and 95% confidence.

Using an MTT assay, the powders and extracts for the liposomes and niosomes of *Synechocystis* sp., in five concentrations (100 μg, 25 μg, 6.3 μg, 1.6 μg, and 0.4 μg) replicated three times and incubated for 48 h at 37 °C, with doxorubicin as a positive control, were compared against cancer cell lines, specifically against Caco2, OVCAR4, and MCF7, for their effect on non-cancerous cell lines, specifically on FHC, OCE1, and MCF10a normal cell lines, so to determine whether the toxicity was selective or if there was a general toxicity that could also cause cytotoxicity to the normal cells.

Cytotoxicity against the cancer cell lines (Caco2, OVCAR4, and MCF7) was higher than against the normal cell lines (FHC, OCE1, and MCF10a). However, the degree of selectivity varied (Table 3).

Meanwhile, the value of cytotoxic selectivity can be determined by calculating the Selectivity Index (SI) value. The acceptable value for the selectivity index was greater than one, which indicates that the drug has more toxicity to cancer cells than to normal cells. This means that selectivity to cancer cells increases as the SI value increases [113,114]. Each SI value was calculated in Table 3 using the following formula:$$SI=\frac{IC50\;for\;normal\;cells}{IC50\;for\;cancer\;cells}$$

Besides the dose–response curves, the results showing L1, L2, N1, and N2, and doxorubicin as a positive control, terminating in the Caco2, OVCAR4, and MCF7 cancer cell lines and the non-cancerous cell lines FHC, OCE1, and MCF10a using five concentrations (100 μg, 25 μg, 6.3 μg, 1.6 μg, and 0.4 μg) are shown in Figure 8.

### 3.4. Molecular Docking Studies of Isolated Compounds toward EGFR

Molecular docking is an effective, sensitive, and cheap method for drug layout and testing [115,116]. The epidermal growth factor receptor (EGFR) and its downstream signaling pathways are involved in the development and progression of several human tumors. Based on the results of previous studies on the anticancer activities of flavonoid and phenolic compounds on the inhibition of the EGFR as the kinase pathway [115,117,118,119,120,121,122], we chose the EGFR as the tested protein in the molecular docking study.

The achieved cytotoxic activity of the crude extract needed to be explained on a molecular level. Therefore, the dereplicated compounds were in silico evaluated against the epidermal growth factor receptor tyrosine kinase (EGFR). The EGFR normally regulates cell proliferation and is found in the cell membrane. However, its overexpression was reported in many types of cancers, such as lung, breast, and kidney carcinomas [123]. Structurally, the EGFR consists of the following three domains: extracellular receptor-like, trans-membranal, and intracellular kinase domains. The EGFR is established to be in a monomeric form during its resting state, which, upon activation, Lys721 forms an ion pair with the conserved Glu738 to interact with the ATP phosphate groups [124,125]. Its ATP-binding domain in the intracellular region possesses a conserved amino acid sequence with 39 residues located near the ATP binding site, among which Leu718, Val726, Ala743, Met793, and Leu844 showed abundant ligand interaction [126].

An examination of the binding interactions of Erlotinib to the active site of the enzyme shows strong hydrogen bond interactions with MET769, CYS773, GLN767, and Met487. The results are summarized in Appendix A and Figure 9.

## 4. Discussion

Based on the results of the characterization of the nanoparticle vesicles studied, the results in Table 2 and the visual examination indicate that the formulated NPs are physiochemically good, have a high stability, and exhibit an anionic surface charge+. The organoleptic properties have a clear appearance with no agglomerated or separate layers depending on sight. The physiological pH at which the liposomes are stable is considered close to a pH of 7.4 [110]. Regarding the stability, according to the results listed in Table 2, the niosomes (N1 and N2) have a zeta potential lower than −30 mV, which is expected to possess enough physical stability [127]. In addition, N1 and N2 had PDI values of 0.31 and 0.35, respectively, where lower PDI values (≤0.3) correspond to a more uniform homogenous dispersion [128]. This slight increase in PDI values returns the niosomes to the hydration process, but still with an accepted range that, in some cases, may extend to seven-tenths but will not exceed one [129]. In the same table, the liposomes’ physical stability returned to higher levels of zeta potential values (0.43.7 and 31.6 mV) and normal PDI values (0.27 mV and 0.24 mV), as also reported in the literature [127,128,129].

Regarding cellular uptake, it is fortunate that the large surface area of the formulated niosomes and liposomes induces slow cellular interaction and low membrane binding, but higher cellular internalization (4).

Additionally, the size of the NPs varies over a wide range, ranging from 1 to 1000 nanometers, depending on the number of bilayers and the nature of the particles, the presence of fibers in the loaded drug, and the formulation method [128,130].

According to the results of the MTT examination and the calculated value of the selectivity index (SI) [113,114] in Table 3, the L1 and L2 results indicated that the strongest activity appeared in the breast (MCF7) cancer cell line, with an acceptable SI value. Despite the L1 and L2 results in the colon (Caco2) and ovarian (OVCAR4) cancer cell lines, they have shown somewhat neutral results in terms of efficacy and selective cytotoxicity. When considering the results of the nano-niosomes, N1 has an impressive result. It recorded the highest effectiveness against the breast (MCF7) cancer cell line, with a very high SI value, thus outperforming doxorubicin. Moreover, it achieved good effectiveness in the colon (Caco2) and ovarian (OVCAR4) cancer cell lines, with great SI values, which makes it subject to upcoming study. Additionally, N2 achieved the highest effect against colon cancer cell lines.

In general, nano-niosomes were more effective than nano-liposomes. The effectiveness of nano-niosomes may be due to their composition. They consist of nonionic surfactants that are less susceptible to degradation and are more stable. This allows them to deliver the drug to the target organ and increases their cytotoxic activity. In contrast, nano-liposomes consist of phospholipids, which are considered less stable and quicker to degrade [30].

Additionally, using the dry powder directly showed greater effectiveness in all of the formulations compared to the ethanolic extract. In general, the organic solvent increased the protein content of the extract compounds and successfully extracted some active compounds [131,132]. Its disadvantages include lower selectivity, which blocks many insoluble dyes and other substances that could participate in the required bioactivity [132,133]. The dry powder retained almost all of the proportions of the natural compounds, highlighting the possibility that all of the compounds have the same potential for biological effectiveness, encouraging the reduction in the use of organic solvents.

In reviewing the previous literature on other nano-formulations and their effects on the cancer cells under study, many silver nanoparticles have been biosynthesized using different organisms and have been studied on the MCF-7 human breast cancer cell line [134]. Nickel and copper nanoparticles have also been biosynthesized and studied on anti-human ovarian cancer [135,136], while other nanoparticles have been studied on cancer cells [137,138]. By comparing the cytotoxicity results in Table 3 with the cytotoxicity results in previous studies, it is found that all of the formulations in Table 3 showed superiority to most other formulations in previous studies, especially N1, which confirms a new strategy for combating cancer.

Furthermore, the compounds identified through the LC-MS and metabolite identification, as shown in Table 1, have shown different biological activities, particularly when focusing on their activity against cancer. Both oleic acid and Scytonemin have shown the ability to directly stimulate the growth of cancer cells through apoptosis, which has multiple mechanisms of action, including stopping the cell cycle and inducing programmed cell death [139,140]. β-Carotene may suppress cancer indirectly through its ability to regulate the immune system [141]. Abietic acid and Palmitoleic acid also have promising results in fighting cancer and stimulating cell death, but their mechanisms of action are not fully understood [93,142]. The same is the case in glycolipid PF2 and medusamide A [86,87,97]. Although zeaxanthin does not affect cancer cells directly, it can contribute to their elimination through antioxidant and anti-inflammatory properties [143]. This confirms the *Synechocystis* sp. extract’s effectiveness as a promising natural preparation for cancer treatment.

Consequently, the observed cytotoxic effects can be attributed to several mechanisms related to the bioactive compounds identified in the *Synechocystis* sp. extract. Nano-formulations improve the delivery of bioactive compounds to cancer cells by increasing their solubility and stability, ensuring that higher concentrations reach the target site. The nano-sized particles facilitate easier penetration into cancer cells, allowing for the more effective intracellular delivery and action of the bioactive compounds. Liposomes and niosomes provide a sustained release of the encapsulated compounds, maintaining therapeutic levels for extended periods and improving the overall cytotoxic efficacy.

The high Selectivity Index (SI) values observed for the nano-formulations, particularly N1 (niosomes), against the MCF7 (breast cancer) cell line indicate that the nano-formulations significantly enhance the selectivity and potency of the bioactive compounds, reducing their cytotoxic effects on normal cells. Hence, the ideal antitumor drug cannot be determined solely by the factor that destroys cancer cells, but must also be safe, have a selectivity of cytotoxic activity, and produce a maximum effect at the lowest dose with minimum side-effects [144].

Additionally, the results of the molecular docking studies of the isolated compounds toward the EGFR are found in Appendix A and Figure 9. Compound **21** showed the highest binding energy score (−9.5 kcal/mol) among the tested compounds, indicating a higher fitting ability, followed by compounds **6**, **7**, **8**, **12**, and **19** (−9.2 kcal/mol). Moreover, compounds **6**, **7**, **8**, **12**, **19**, and **21** managed to form hydrogen bonds with the crucial Lys721. Consequently, as explained earlier, the achieved binding to Lys721 might hinder its function in activating the EGFR. Also, compound **21** formed a hydrogen bond with Glu 738. Furthermore, compounds **1**, **2**, **3**, **6**, **14**, and **22** interacted with Asp831 to form H-bonds.

The 2D interaction diagrams (Figure 9) illustrate the specific interactions between the compounds and the EGFR binding site. These visualizations reveal crucial binding sites and bond types, highlighting the potential of these compounds to inhibit EGFR activity.

The interaction diagrams in Figure 9 provide a detailed view of the docking poses and key residues involved in binding. For instance, the strong hydrogen bond interactions with Lys721 observed in compounds **6**, **7**, **8**, **12**, **19**, and **21** suggest their potential to block the ATP-binding site, which is crucial for kinase activity inhibition.

The observed cytotoxic effects of the compounds are likely due to their effective binding and inhibition of the EGFR. The strong interaction with key residues, particularly Lys721 and Glu738, suggests that these compounds can significantly disrupt EGFR signaling pathways. This disruption is critical in inhibiting cancer cell proliferation, as evidenced by the high binding affinities and observed cytotoxicity.

The detailed docking scores and interaction analyses enhance the understanding of how these compounds exert their cytotoxic effects, supporting their potential as effective EGFR inhibitors, as shown in (Table 4).

The molecular docking studies identified compound **21** as having the highest affinity for the EGFR, with a binding energy score of −9.5 kcal/mol. This high binding affinity suggests a strong potential for inhibiting EGFR activity. Other notable compounds include compounds **6**, **7**, **8**, **12**, and **19**, each demonstrating significant binding energies of −9.2 kcal/mol. These compounds effectively interacted with key residues, particularly Lys721 and Glu738, which are crucial for the EGFR’s ATP-binding function. The strong binding interactions suggest that these compounds could significantly disrupt EGFR signaling pathways, thereby inhibiting cancer cell proliferation.

As described above, the research results obtained confirm that the production of nano-liposomes and nano-niosomes using *Synechocystis* sp. is a promising system against cancer, as *Synechocystis* sp. has proven to possess effective compounds that can fight cancer through the programmed death of cancer cells, damage to their DNA, and other effects that lead to the elimination of cancer cells. In addition to identifying the compounds that have been studied through molecular docking studies, they can deter cancer by various mechanisms of action that include preventing the production of cancer cells or controlling the cell membrane, etc. In short, the following new field should be highlighted: the production of anticancer drugs by *Synechocystis* sp., which can be said to be a green treasure.

## 5. Conclusions

This study showed, for the first time, the effectiveness of the cyanobacteria strain *Synechocystis* sp. to synthesize nano-liposomes and nano-niosomes that fight against different cancer cell lines, specifically the colon, ovary, and breast cancer cell lines Caco2, OVCAR4, and MCF7, respectively, in comparison to normal cell lines (FHC, OCE1, and MCF10a). N1 recorded the highest effectiveness against the breast cancer cell line (MCF7), with an IC50 value of 2.38 µg/mL. The nano-niosome formulations, in general, also showed the highest efficacy against all cancer lines. Furthermore, in all formulations, the dry powder was more effective than the ethanol extract, which encourages the reduction in the use of organic solvents in nano-formulations and thus avoiding their environmental harm. These results encourage researchers to focus on cyanobacteria in general, and especially *Synechocystis* sp., for the synthesis of new drugs to fight against human cancers, with fewer side-effects and greater effectiveness. In addition, synthesizing nano-liposomes and nano-niosomes by cyanobacterial extracts is a relatively new idea that can achieve promising results and will be possible in developing medicines. Furthermore, molecular docking identified several compounds with promising interaction potential. These compounds, including 1-hexadecanoyl-2-(9Z-hexadecenoyl)-3-(6′-sulfo-alpha-D-quinovosyl)-sn-glycerol, Sulfoquinovosyl monoacylgycerol Scytonemin, 3-Hydroxymyristic acid, Glycolipid PF2, Palmitoleic acid, and Glyceryl monostearate, formed hydrogen bonds with key residues (Lys721 and Asp831) within the enzyme’s active site. This suggests that they warrant further investigation through isolation and dedicated studies so to understand their biological effects. The results indicate a new potential strategy for combating cancer and highlight the presence of anticancer compounds in the *Synechocystis* sp. extract. This discovery offers a promising avenue for therapeutic development and warrants further investigation of these formulations, progressing from laboratory research to clinical trials. However, it is crucial to acknowledge that in vitro assays may not consistently reflect compound effects in vivo, necessitating additional evaluation in animal models and clinical trials to assess the safety and efficacy of potential drugs. Various experimental challenges must be addressed, including storage conditions, production scalability, customized dosage formulations, and safety considerations. Future research endeavors could focus on developing antitumor nano-liposomes and nano-niosomes utilizing *Synechocystis* sp. extract, transitioning them to clinical trials, and addressing potential challenges in their implementation.

## Figures and Tables

**Figure 1 biology-13-00581-f001:**
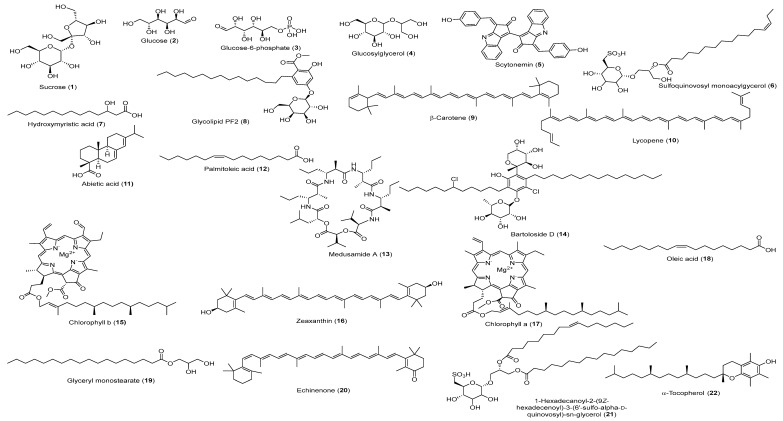
The chemical structures of tentatively identified compounds from the *Synechocystis* sp. extract.

**Figure 2 biology-13-00581-f002:**
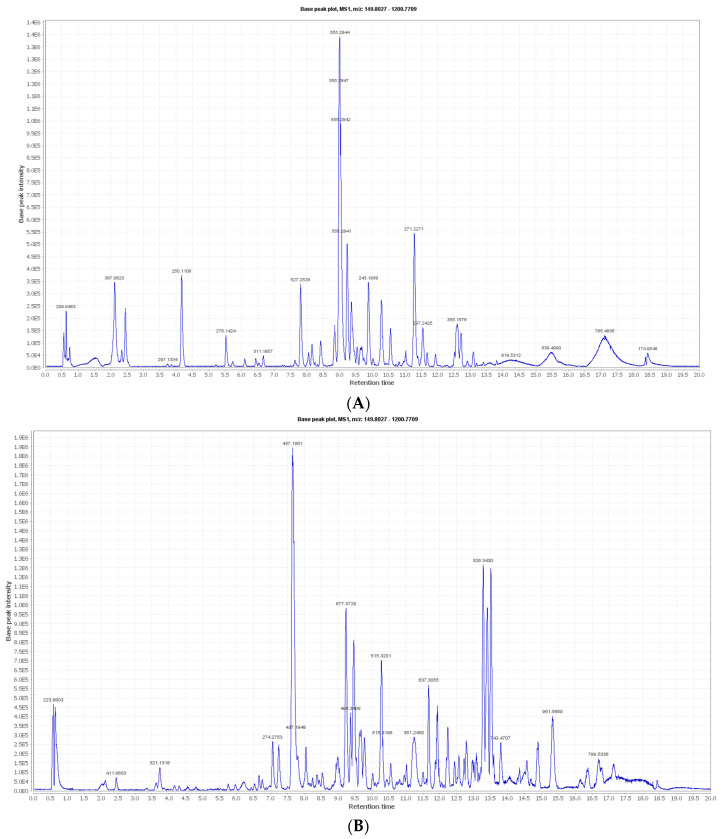
Total ion chromatograms (TICs) of the *Synechocystis* sp. extract in (**A**) negative and (**B**) positive modes following the analysis by liquid chromatography–high-resolution electrospray ionization mass spectrometry (LC-HRESI-MS).

**Figure 3 biology-13-00581-f003:**
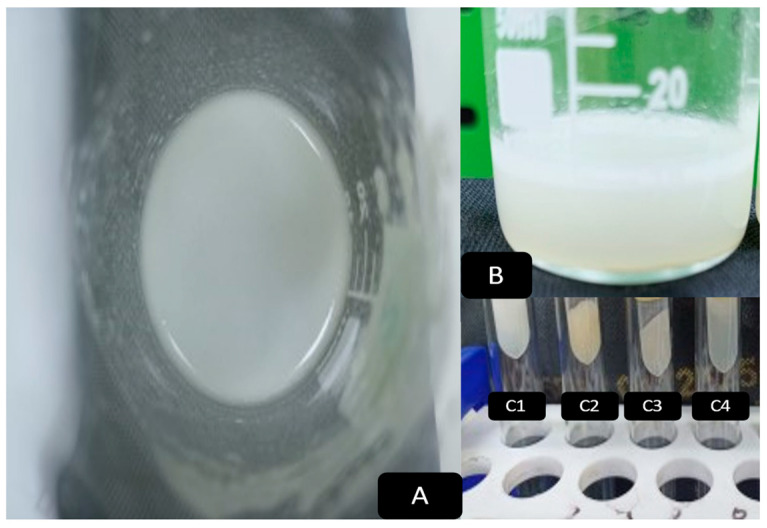
(**A**) Top view of the prepared vesicles; (**B**) Side view of the prepared vesicles; (**C1**) L1 preparation; (**C2**) L2 preparation; (**C3**) N1 preparation; (**C4**) N2 preparation.

**Figure 4 biology-13-00581-f004:**
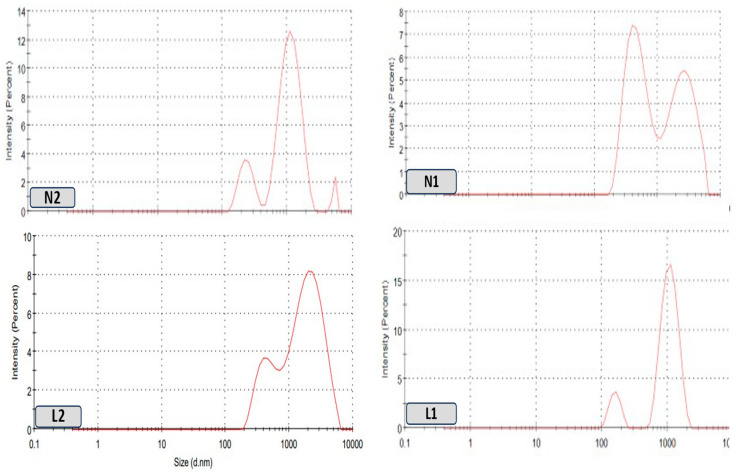
Size distributions of the different vesicles, L1, L2, N1, and N2, of the dispersion versus doxorubicin as a positive control.

**Figure 5 biology-13-00581-f005:**
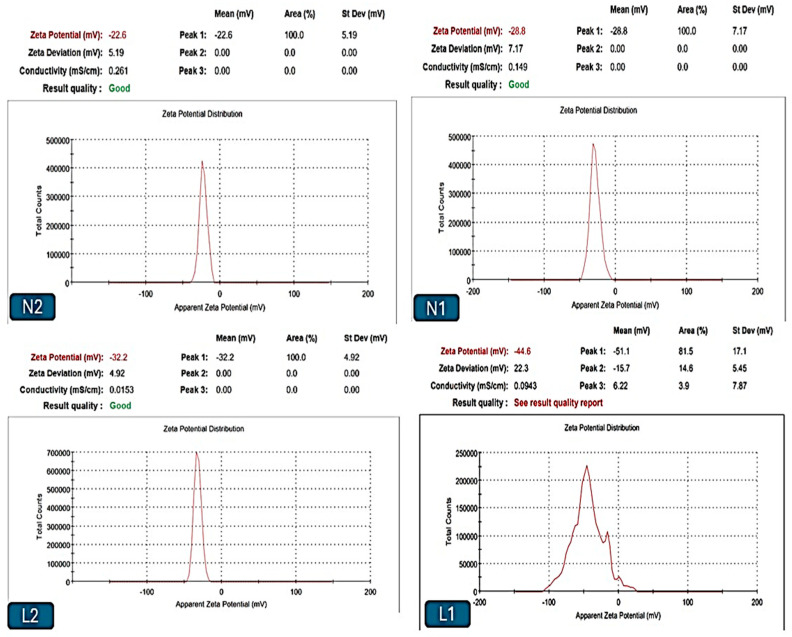
Zeta potentials of the different vesicles, L1, L2, N1, and N2.

**Figure 6 biology-13-00581-f006:**
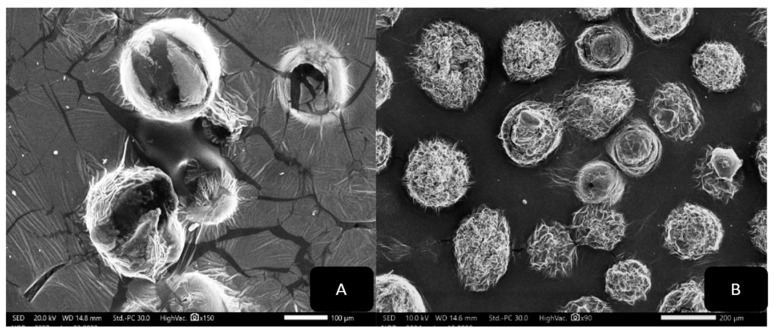
(**A**) SEM scanning of L1; (**B**) SEM scanning of L2.

**Figure 7 biology-13-00581-f007:**
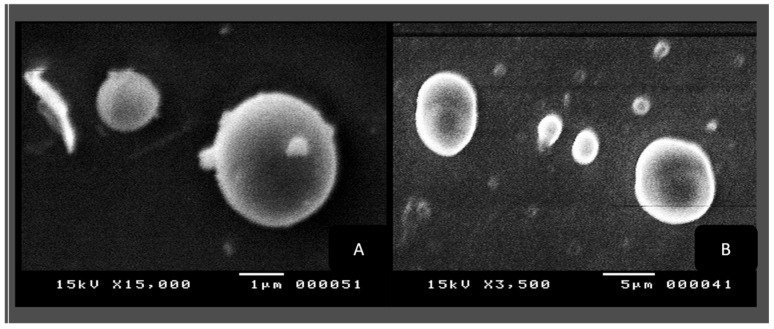
(**A**) SEM scanning of N1; (**B**) SEM scanning of N2.

**Figure 8 biology-13-00581-f008:**
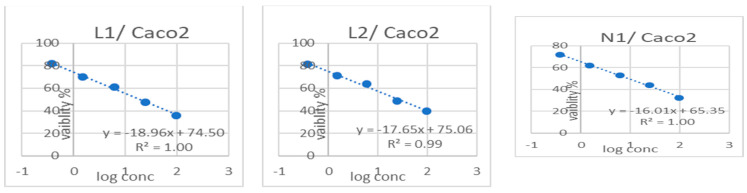
Dose–response curves for L1, L2, N1, and N2, and doxorubicin as a positive control, terminated in Caco2, OVCAR4, MCF7 cancer cell and FHC, OCE1, and MCF10a non-cancerous cell lines, with 5 concentrations (100 μg, 25 μg, 6.3 μg, 1.6 μg, and 0.4 μg); mean ± SD of 3 independent experiments.

**Figure 9 biology-13-00581-f009:**
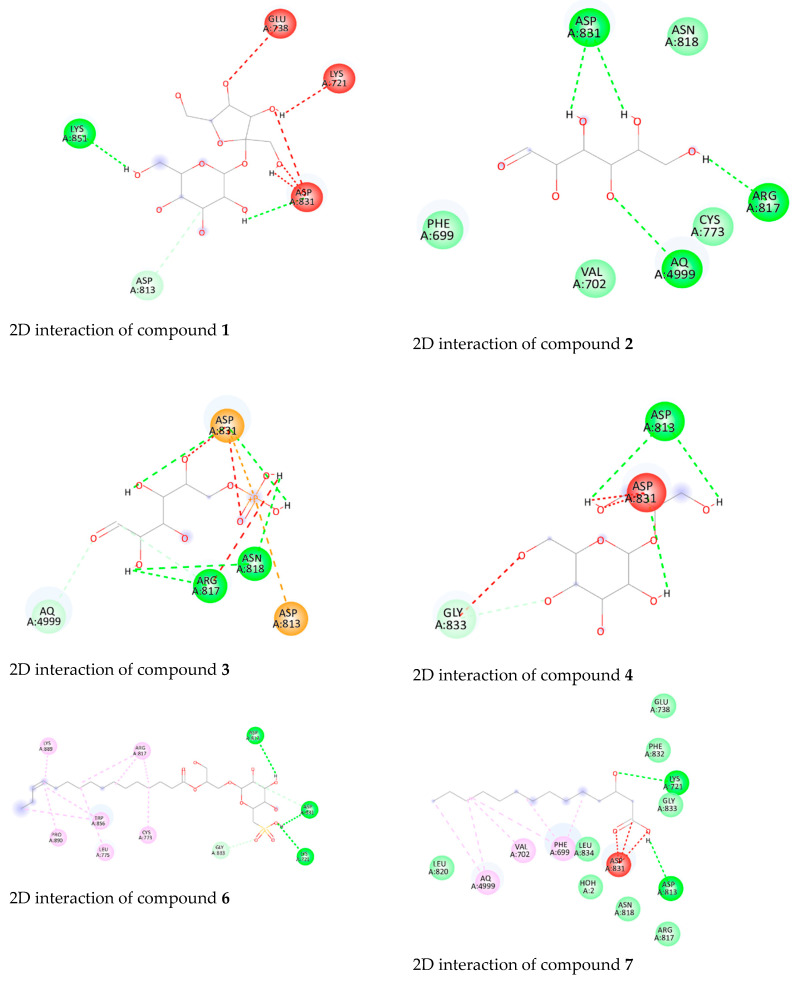
2D interaction of metabolites identified in the *Synechocystis* sp. extract (compounds **1**–**14**, **16**, and **18**–**22**) with the active site of EGFR binding.

**Table 1 biology-13-00581-t001:** List of tentatively identified compounds from *Synechocystis* sp. extracts by liquid chromatography–high-resolution electrospray ionization mass spectrometry (LC-HRESI-MS) following the previous literature.

Identified Compound	Molecular Formula	Phytochemical Class	*m*/*z*	RT (min)	M wt.	LM1.Mzxml Peak Area	Blank_Neg.Mzxml Peak Area	Exact Mass Difference	Potential Bioactivities	Source	Ref.
Sucrose	C_12_H_22_O_11_	Sugars	341.1076	0.652017	342.1149	0	0	1	Not reported	*Synechocystis* 7338 and *Synechocystis* 6803	[77]
Glucose	C_6_H_12_O_6_	Sugars	179.0545	2.326483	180.0618	0	0	1	Not reported	*Synechocystis* sp.	[77]
Glucose-6-phosphate	C_6_H_13_O_9_P	Sugars	259.0235	3.106233	260.0308	0	0	1	Not reported	*Synechocystis* sp.	[77]
Glucosyl glycerol	C_9_H_18_O_8_	Sugars	253.1047	3.679633	254.112	0	0	10	Not reported	*Synechocystis* sp.	[77]
Scytonemin	C_36_H_20_N_2_O_4_	Phenolic pigments	543.1698	3.708917	544.1771	0	0	10	Photoprotective; anticancer agent	*Synechocystis* sp.	[78,79,80,81]
Sulfoquinovosyl monoacylgycerol	C_25_H_46_O_11_S	Sulfolipids	553.2688	8.159594	554.2761	383,504	0	0	Antiviral effect	*Synechocystis* sp.	[82,83]
3-Hydroxymyristic acid	C_14_H_28_O_3_	Hydroxy fatty acid	243.1959	8.872878	244.2031	1,436,656	2610.386	0	Not reported	*Synechocystis* sp.	[84]
Glycolipid PF2	C_29_H_48_O_9_	Glycolipids	541.3353	10.54202	540.328	434,467.4	0	0	Anti-inflammatory, anticancer	*Synechocystis* sp. PCC	[85,86,87]
*β*-Carotene	C_40_H_56_	Carotenoids	537.4475	10.73032	536.4402	0	0	5	Neuroprotective, antioxidant, anticancer	All Cyanobacterial species	[76,78,88,89,90]
Lycopene	C_40_H_56_	Carotenoids	537.4475	10.73032	536.4402	0	0	2	Anticarcinogenic, antiatherogenic	*Synechocystis salina*	[90,91]
Abietic acid	C_20_H_30_O_2_	Diterpenoids	303.232	11.02424	302.2247	0	411,183.5	0.2	Anticancer	*Synechocystis* sp.	[92,93]
Palmitoleic acid	C_16_H_30_O_2_	Fatty acids	253.2162	11.61243	254.2235	4613.705	0	1	Anticancer	*Synechocystis* 6803	[94,95]
Medusamide A	C_44_H_79_N_5_O_9_	Depsipeptides	822.5954	11.67082	821.5881	1798.30	0	4	Anticancer	*Synechocystis* sp.	[96,97]
Bartoloside D	C_44_H_75_Cl_3_O_10_	Alkylresorcinols	869.4769	11.73558	870.4842	13,325.94	0	0	Not reported	*Synechocystis salina*	[98]
Chlorophyll b	C_55_H_70_MgN_4_O_6_	Chlorophyll	905.513	12.0794	906.5203	0	0	5	Not reported	All Cyanobacrerial sp.	[99,100]
Zeaxanthin	C_40_H_56_O_2_	Xanthophylls	569.4195	12.31082	568.4122	0	0	10	Anticancer	*Synechocystis salina*	[94,101,102]
Chlorophyll a	C_55_H_72_O_5_N_4_Mg	Chlorophyll	891.5305	12.34362	892.5377	0	0	2	Neuroprotective	All *Cyanobacterium* sp.	[88,99,100]
Oleic acid	C_18_H_34_O_2_	Fatty acids	281.2473	12.69699	282.2546	411,919.7	0	1	Anticancer	*Synechocystis* 6803	[94,103,104]
Glyceryl monostearate	C_21_H_42_O_4_	Glycerides	357.2998	12.90373	358.3071	93,861.73	0	1	Antioxidant	*Synechocystis* 7338/*Synechocystis* 6803	[77,105]
Echinenone	C_40_H_54_O	Carotenoids	551.4238	13.09513	550.4165	0	0	0.9	Anti-Alzheimer	*Synechocystis salina*	[106,107]
1-hexadecanoyl-2-(9Z-hexadecenoyl)-3-(6′-sulfo-alpha-D-quinovosyl)-sn-glycerol	C_41_H_76_O_12_S	Sulfolipids	791.4996	17.22822	792.5069	3,203,772	0	0	Not reported	*Synechocystis* sp. PCC 6803	[108]
α-Tocopherol	C_29_H_50_O_2_	Vitamins	431.3854	17.65833	430.3782	0	0	2	Antioxidant, antimicrobial	*Synechocystis* 6803	[94,109]

**Table 2 biology-13-00581-t002:** Different characteristics of the prepared vesicle dispersions.

Formulation Codes	Surfactant Ratio Ch:Lec *	Sample Type	Amount (mL)	Size (nm)	PDI	Zeta Potential (mV)
L1 *	0:1	Emulsion	0.5	419 ± 114	0.27 ± 0.32	−43.7 ± 1.65
N1 *	1:1	Emulsion		541 ± 479.4	0.31 ± 0.1	−28.7 ± 1.3
L2 *	0:1	Emulsion		847 ± 173.2	0.24 ± 0.22	−31.6 ± 1.29
N2 *	1:1	Emulsion		1051 ± 479.4	0.35 ± 0.33	−22.2 ± 1.8

L1 * = powder liposomes, N1 * = powder niosomes, L2 * = extract liposomes, N2 * = extract niosomes, Ch:Lec * = cholesterol to lecithin ratio.

**Table 3 biology-13-00581-t003:** Cytotoxicity results of the cancer cell lines and normal cell lines of the powders and extracts for the nano-liposomes and nano-niosomes of *Synechocystis* sp. and doxorubicin as a positive control, SD ± (*n* = 3), as well as the SI values. Statistical analysis was conducted using a one-way ANOVA; groups with different letters are statistically different at *p* ≤ 0.05 according to Tukey’s HSD test (95% CIs).

Sample Code	Cytotoxicity IC50 (µg/mL)						(SI)		
	FHC	Caco2	OCE1	OVCAR4	MCF10a	MCF7	FHC/Caco2	OCE1/OVCAR4	MCF10a/MCF7
L1	39.029 ± 1.36 ^b^	19.56 ± 0.72 ^b^	42.228 ± 2.5 ^b^	33.52 ± 1.55 ^b^	27.177 ± 1.12 ^c^	9.24 ± 0.41 ^c^	1.99	1.25	2.94
L2	65.135 ± 2.27 ^a^	26.27 ± 0.94 ^a^	87.903 ± 5.21 ^a^	56.23 ± 2.38 ^a^	51.169 ± 2.11 ^a^	19.61 ± 0.78 ^b^	2.47	1.5	2.60
N1	29.637 ± 1.04 ^c^	9.09 ± 0.33 ^d^	29.911 ± 1.77 ^c^	11.42 ± 0.72 ^d^	20.135 ± 0.83 ^d^	2.38 ± 0.13 ^e^	3.26	2.61	8.46
N2	26.724 ± 0.93 ^c^	15.57 ± 0.81 ^c^	26.651 ± 1.58 ^c^	18.17 ± 0.61 ^c^	41.865 ± 1.72 ^b^	35.31 ± 1.29 ^a^	1.7	1.46	1.18
Doxorubicin	12.982 ± 0.45 ^d^	3.61 ± 0.26 ^e^	14.157 ± 0.84 ^d^	9.41 ± 0.39 ^d^	25.36 ± 1.04 ^c^	5.35 ± 0.22 ^d^	3.59	1.50	4.74

**Table 4 biology-13-00581-t004:** Molecular docking results of isolated compounds toward the EGFR.

Compound	Binding Energy (kcal/mol)	Key Interactions
**1**	−8.7	Asp831 (H-bond)
**2**	−8.5	Asp831 (H-bond)
**3**	−8.9	Asp831 (H-bond),
		Arg817 (H-bond)
**6**	−9.2	Lys721 (H-bond),
		Asp831 (H-bond)
**7**	−9.2	Lys721 (H-bond),
		Asp831 (H-bond)
**8**	−9.2	Lys721 (H-bond),
		Glu738 (H-bond)
**12**	−9.2	Lys721 (H-bond)
**14**	−8.8	Asp831 (H-bond)
**19**	−9.2	Lys721 (H-bond),
		Glu738 (H-bond)
**21**	−9.5	Lys721 (H-bond),
		Glu738 (H-bond)
**22**	−8.8	Asp831 (H-bond)

## Data Availability

All data generated or analyzed during this study are included in this published article (and its Appendix A).

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
