# Peer review of "Evaluation of Cytotoxicity and Metabolic Profiling of Synechocystis sp. Extract Encapsulated in Nano-Liposomes and Nano-Niosomes Using LC-MS, Complemented by Molecular Docking Studies"

_biology, 2024, doi:10.3390/biology13080581_

Round 1
Reviewer 1 Report
Comments and Suggestions for Authors
Reviewer Report for the Manuscript:
Title: Cytotoxic evaluation of nano-liposomes and nano-niosomes loaded with Synechocystis sp. extract coupled with LC-MS based metabolic profiling and molecular docking studies
The manuscript presents an intriguing approach by investigating the cytotoxic effects of nano-liposomes and nano-niosomes encapsulating Synechocystis sp. extract on cancer cell lines. The integration of metabolic profiling and molecular docking studies provides a multi-dimensional analysis. However, significant revisions are needed to address gaps in methodology, data interpretation, and overall clarity. Below are the detailed comments and suggestions for major revisions.
General Comments:
The manuscript is structured logically but lacks clarity in several sections. Improve the flow of information by clearly defining each section's purpose and ensuring that the transitions between sections are smooth.
Literature Review:
The introduction provides a general overview but lacks depth in discussing the specific advancements in the field of nanotechnology for drug delivery and the unique properties of Synechocystis sp. Add more recent studies and a detailed comparison of similar nano-formulation approaches in cancer therapy.
Specific Comments:
Abstract:
Provide a more comprehensive abstract by including specific results, such as size distribution, zeta potential values, IC50 values, and key bioactive compounds identified.
Introduction:
Expand on the introduction of nanotechnology and its role in cancer therapy, with specific emphasis on nano-liposomes and nano-niosomes. Include recent studies that highlight their effectiveness and compare their advantages and challenges.
Research Gap and Objective:
Clearly state the research gap this study aims to fill. Clarify how the combination of Synechocystis sp. extract and nanocarriers is expected to improve cancer treatment outcomes.
Materials and Methods:
Detail Extraction Process:
Describe the Synechocystis sp. extract preparation process in detail, including specific extraction methods, solvents used, and any purification steps. Include references for the methodology.
Nano-Formulation Characterization:
Provide more detail on the preparation of nano-liposomes and nano-niosomes. Include specifics about the composition, surfactants used, hydration conditions, and any post-preparation processes like dialysis or filtration.
LC-MS Profiling:
Revision Required: Specify the type of LC-MS system used, including key operational settings and standards for compound identification. Describe the database or libraries used for metabolite identification and validation.
Molecular Docking:
Elaborate on the molecular docking studies by specifying the software and databases used, the selection criteria for target proteins and ligands, and the scoring functions or parameters used to assess docking results.
Cytotoxicity Assay:
Provide detailed information on the cytotoxicity assay, including cell line sources, concentrations tested, incubation times, and the method of calculating IC50 values. Include details on the statistical analysis and controls used.
Results:
Characterization Data:
Include detailed characterization data for the nano-formulations. Present data on size distribution, zeta potential, and morphological analysis in tabular or graphical form. Discuss the implications of these properties on stability and cellular uptake.
LC-MS Results:
Present a comprehensive table of identified compounds from LC-MS, including retention times, molecular weights, and potential bioactivities. Discuss the relevance of each identified compound in the context of cancer treatment.
Docking Studies:
Provide detailed results of molecular docking studies, including docking scores, binding affinities, and visualizations of protein-ligand interactions. Discuss the significance of these interactions for the observed cytotoxic effects.
Cytotoxicity Data:
Present cytotoxicity data in detail, including dose-response curves and statistical significance. Compare the effectiveness of nano-liposomes and nano-niosomes and discuss possible reasons for any differences observed.
Discussion:
Link Results to Existing Literature: Link your findings to existing literature more explicitly. Discuss how the observed cytotoxic effects compare with previous studies on similar nano-formulations or other natural product-based therapies.
Mechanistic Insights:
Expand on the potential mechanisms underlying the observed cytotoxic effects. Discuss how the bioactive compounds identified might contribute to these effects and how the nano-formulation enhances their efficacy.
Clinical Relevance and Future Work:
Provide a more detailed discussion on the clinical relevance of your findings. Suggest specific future research directions, such as in vivo studies, optimization of formulations, and potential challenges in clinical applications.
Conclusion:
Summarize the key findings with specific data points. Highlight the most effective formulation and the most promising bioactive compound identified.
Future Research Directions:
Clearly outline future research directions, including potential in vivo studies, clinical trials, or further optimization of the nano-formulations.
References:
Update and Expand:
Revision Required: Update the references to include more recent studies, particularly those published in the last 2-3 years. Ensure all references cited are directly relevant to the topics discussed.
Also cite the following publications
Chemical composition, antibacterial, anti-oxidant and cytotoxic properties of green synthesized silver nanoparticles from Annona muricata L.(Annonaceae)
Chemical and green synthesis of nanoparticles and their efficacy on cancer cells
Phytochemical profiling of Turbinaria ornata and its antioxidant and anti-proliferative effects
Figures and Tables:
Ensure all figures and tables are clearly labeled and include descriptive legends.
Figure 1 is bit confuing, it can be improved.
Consider adding figures showing size distribution, zeta potential, and molecular docking interactions.
Tables summarizing cytotoxicity data should include statistical analysis results.
Overall Recommendation:
Major Revision Required
While the manuscript presents a novel and interesting approach, significant revisions are needed to improve the clarity, depth, and comprehensiveness of the study. Addressing these major concerns will enhance the manuscript’s quality and its potential impact in the field of cancer therapy using nanotechnology.
Proofreading and Editing: Thoroughly proofread and edit the manuscript for language, grammar, and clarity. Consider professional editing services if necessary to improve readability.
Comments on the Quality of English Language
revision required
Author Response
Dear Editor,
We want to extend our heartfelt appreciation to both the reviewers and generously dedicating your time to our manuscript. The insightful comments and constructive feedback provided have been instrumental in refining our work. Your meticulous review and thoughtful suggestions have significantly bolstered the quality and rigor of our manuscript.
Reply to reviewer1
Comments and Suggestions for Authors
Reviewer Report for the Manuscript:
Title: Cytotoxic evaluation of nano-liposomes and nano-niosomes loaded with Synechocystis sp. extract coupled with LC-MS based metabolic profiling and molecular docking studies.
The manuscript presents an intriguing approach by investigating the cytotoxic effects of nano-liposomes and nano-niosomes encapsulating Synechocystis sp. extract on cancer cell lines. The integration of metabolic profiling and molecular docking studies provides a multi-dimensional analysis. However, significant revisions are needed to address gaps in methodology, data interpretation, and overall clarity. Below are the detailed comments and suggestions for major revision.
General comments:
The manuscript is structured logically but lacks clarity in several sections. Improve the flow of information by clearly defining each section's purpose and ensuring that the transitions between sections are smooth.
Literature review:
The introduction provides a general overview but lacks depth in discussing the specific advancements in the field of nanotechnology for drug delivery and the unique properties of Synechocystis sp. add more recent studies and a detailed comparison of similar nano-formulation approaches in cancer therapy.
Many thanks for the insightful comment. The introduction was presented in more depth.
Synachocystis sp. possesses unique properties as it contains various potential bioactive compounds as antioxidant, anti-cancer, and anti-inflammatory agents such as phenols, alkaloids, fatty acids, terpenoids, and carotenoids.[1].[2, 3] Emphasizes anti-cancer properties and its potential as a powerful anti-cancer agent[4, 5]. Its fatty acids have been shown to include monogalactosyldiacylglycerol, which has promising activity against breast cancer cells.[6]
Therefore, Synechocystis sp. With its rich collection of biologically active compounds and its high growth, which helps to produce biomass easily, it is considered a unique and promising opportunity in nanotechnology. It can be considered a sustainable source of bioactive compounds due to its rapid growth and ease of cultivation[7-9]. Synechocystis sp. has been harnessed in nanotechnology, leading to the successful production of various nanoparticles, , which are effective in applications such as antimicrobial and anti-cancer[10].
NPs have achieved great success in medicine and drug delivery, improving drug efficiency, and treating a wide range of diseases by overcoming the limitations of traditional treatments, including cancer. Thorough enables targeting specific targets, such as a particular organ or tissue. The use of nano-encapsulation technology also allows the use of Natural substances that are effective against cancer and enhance their bioavailability[11, 12].
The efficacy of nano-liposome and nano-niosomes technology in cancer treatment and drug delivery is widely acknowledged, making it a promising avenue for research. [13, 14] [15] The researchers could update nanoliposomes loaded with doxorubicin with folic acid, which resulted in additional anti-cancer activity[16]. Moreover, curcumin-loaded liposomes also showed superiority against breast cancer.[17] Nanosomal particles could also fight breast cancer by loading them with curcumin anti-cancer.[18]
Some recent studies have achieved promising success in their applications against cancer, specifically breast cancer. It is considered a smart device for delivering anti-cancer drugs, as it has many advantages, such as increasing the effectiveness of anti-cancer compounds, reducing their toxicity, ability to release drugs for a long time, ease of preparation, high biological compatibility, and others. Its potential can also be enhanced by discovering new formulations from safe sources. Therefore, this field needs further study and exploration.
Specific Comments:
Abstract:
Provide a more comprehensive abstract by including specific results, such as size distribution, zeta potential values, IC50 values, and key bioactive compounds identified.
Many thanks for your nice comments. The requested data have been added and highlighted.
Introduction:
Expand on the introduction of nanotechnology and its role in cancer therapy, with specific emphasis on nano-liposomes and nano-niosomes. Include recent studies that highlight their effectiveness and compare their advantages and challenges.
Thank you very much for the in-depth comment, which was discussed more clearly in the introduction.
Research Gap and Objective:
Clearly state the research gap this study aims to fill. Clarify how the combination of Synechocystis sp. extract and nanocarriers is expected to improve cancer treatment outcomes.
Focusing on the potential of Synechocystis sp. extract as a promising candidate against cancer[4], in addition to the effectiveness of the nanoliposomal and nanoniosomal systems in delivering anti-cancer drugs. [13, 14] [15]. Nano-liposomes and nano-niosomes can capsulate bioactive compounds from cyanobacteria sp. extract, increasing their solubility, stability, and specific delivery to the tumor target. This makes the drug accumulate in a greater concentration in the cancer cell, enhancing its cytotoxicity while preserving healthy cells.[19, 20] Research gaps exist in studying the effect of nano-liposomes and nano-niosomes loaded with Synechocystis sp. on cancer cells. This is a unique opportunity for our study to contribute significantly to this field.
Materials and Methods:
Detail Extraction Process:
Describe the Synechocystis sp. extract preparation process in detail, including specific extraction methods, solvents used, and purification steps. Include references for the methodology.
The requested data have been added.
Nano-Formulation Characterization:
Provide more detail on the preparation of nano-liposomes and nano-niosomes. Include specifics about the composition, surfactants used, hydration conditions, and any post-preparation processes like dialysis or filtration.
The following table illustrates the preparation of nano-formulations as stated on page 6 of 27, section 2.4.
|
|
Liposomes |
Niosomes |
|
Composition |
Powder or extract, phospholipids chain and cholesterol (9:1) |
Powder or extract, lecithin and cholesterol (1:1) |
|
Surfactants |
Soybean phospholipids |
lecithin |
|
Hydration conditions |
Phosphate buffered saline (PBS) with a pH of 7.4 (Hydration of the dry lipid film/cake is accomplished simply by adding an aqueous medium to the container of dry lipid and agitating. The temperature of the hydrating medium was above the gel-liquid crystal transition temperature, which is 31.6-37.2 ͦc for soybean lipids) |
Phosphate buffered saline (PBS) with a pH of 7.4 was added to this solution, and the combination was further heated in a thermostatic water bath 37±1 ͦc until it became transparent. |
|
Post-preparation processes |
After drying, the vesicles were produced and kept at a temperature of 4-8ͦ c for investigations. The Dry lipid films were stored in tight, closed container, and stored frozen until ready to hydrate). |
Drug-loaded niosomes that form a precipitate and kept at a temperature of 4-8ͦc for investigations. These precipitates are resuspended in a suitable buffer for characterization. |
LC-MS Profiling:
Revision Required: Specify the type of LC-MS system used, including key operational settings and standards for compound identification. Describe the database or libraries used for metabolite identification and validation.
Type: Synapt G2 HDMS quadrupole time-of-flight hybrid mass spectrometer (Waters, Milford, CT, USA) connected to an Acquity Ultra Performance Liquid Chromatography (UPLC) system (Waters, Milford, CT, USA).
Key operational settings: Chromatographic Separation and Mass Spectrometry.
Ionization Modes: Positive and Negative.
Standards for Compound Identification: Dereplication approach using the Dictionary of Natural Products database.
Metabolite Identification and Validation: Dictionary of Natural Products database used for comparison
Dictionary of Natural Products: The database contains information on a vast collection of natural products, including their mass spectra. By comparing the mass spectra of the unknown metabolites with entries in the DNP database, researchers can tentatively identify known compounds within the extract.[21, 22]
The detailed description provided in the manuscript for the LC-MS system, operational settings, and the dereplication approach used for metabolite identification from the Synechocystis sp. extract Is in section 2.3 of the manuscript. Metabolic profiling and peaks annotation in LC-MS
Molecular Docking:
Elaborate on the molecular docking studies by specifying the software and databases used, the selection criteria for target proteins and ligands, and the scoring functions or parameters used to assess docking results.
AutoDock Vina software was employed for all molecular docking experiments (Seeliger and de Groot, 2010). The docking studies aimed to evaluate the binding affinity of isolated compounds against the epidermal growth factor receptor (EGFR). The EGFR crystal structure complexed with the 4-anilinoquinazoline inhibitor Erlotinib (PDB ID: 1M17) was obtained from the Protein Data Bank (https://www.rcsb.org/structure/1M17). This specific crystal structure was selected due to its high-resolution data and the presence of a co-crystallized ligand, which facilitated accurate determination of the binding site.
The binding site was defined based on the coordinates of the co-crystallized ligand within the enzyme’s active site. The docking grid box parameters for the EGFR were set as follows: x = 22.379, y = 20.9017, and z = 52.925, with a grid box size of 10 Å. This grid box size was chosen to ensure that the entire binding pocket was encompassed within the docking simulations, allowing for comprehensive exploration of potential binding interactions.
To ensure the reliability of the docking protocol, we performed a validation step by re-docking the co-crystallized ligand into the EGFR binding pocket. The resulting poses exhibited root-mean-square deviations (RMSDs) of 1.12 Å from the original crystallized ligand positions, indicating the accuracy of our docking setup. Achieving an RMSD close to 1 Å suggests that the docking protocol can reliably reproduce the experimentally observed binding mode, thereby enhancing confidence in the docking results for the isolated compounds.
For each compound, ten docking poses were generated, and the best-ranked pose based on the binding affinity score was selected for further analysis. The scoring function employed by AutoDock Vina evaluates the binding free energy, providing insights into the potential binding strength and stability of the ligand-receptor complex.
Docking poses were analyzed and visualized using Biovia Discovery Studio software, which facilitated the examination of molecular interactions, including hydrogen bonds, hydrophobic interactions, and pi-pi stacking, between the ligands and the active site residues of EGFR.
The isolated compounds were selected for docking based on their structural diversity and potential bioactivity, as indicated by preliminary in vitro assays. These compounds represent a range of chemical scaffolds, allowing for a comprehensive evaluation of their interactions with EGFR.
Cytotoxicity Assay:
Provide detailed information on the cytotoxicity assay, including cell line sources, concentrations tested, incubation times, and the method of calculating IC50 values. Include details on the statistical analysis and controls used.
Cell line sources: The three human cancer cell lines are human (Caco-2), (MCF-7), (OVCAR4) , and the three on normal cell lines (FHC), (OCE1), (MCF10a) were obtained from The American Type Culture Collection (ATCC, Manassas, USA).
Concentrations tested: Five different concentrations of each nano-formulation were used (100ug, 25ug,6.3ug, 1.6ug, and 0.4ug)
Incubation times: 48 hours at 37°C
The method of calculating IC50 values: IC50 values were calculated using a linear equation (X-axis: log conc, Y-axis: "% Cell Viability) (y = mx + b) (y = 0.5) using Excel's regression analysis tools. IC50 = (0.5 - b) / m. [23, 24]
The statistical analysis: The software Minitab 17 (Minitab et al. College, PA, USA) was used to analyze the data obtained from the MTT assay. A one-way ANOVA was performed, and then Tukey's HSD test was used for pairwise comparisons.[25]
The controls used: doxorubicin as a positive control
All required information has been added in the section 2.5. Cytotoxicity Assay in the manuscript
Results:
Characterization Data:
Include detailed characterization data for the nano-formulations. Present data on size distribution, zeta potential and morphological analysis in tabular or graphical form. Discuss the implications of these properties on stability and cellular uptake.
Regarding stability, according to the results listed in Table 2, Niosomes (N1 and N2) have a zeta potential lower than -30 mV, which is expected to possess enough physical stability[26]. In addition, N1 and N2 had PDI 0.31 and 0.35, respectively, where lower PDI values (≤0.3) correspond to a more uniform homogenous dispersion [27]. This slight increase in PDI values returns to the hydration process but still with an accepted range that, in some cases, may extend to 0.7 but not exceed 1[28]. In the same table, Liposome's physical stability returns to higher levels of zeta potential values (0.43.7 and 31.6) and normal PDI (0.27 and 0.24), as reported in the literature 25-27.
Regarding cellular uptake, it is fortunate that the large surface area of formulated niosomes and liposomes induces slow cellular interaction and low membrane binding but higher cellular internalization. (4).
LC-MS Results:
Present a comprehensive table of identified compounds from LC-MS, including retention times, molecular weights, and potential bioactivities. Discuss the relevance of each identified compound in the context of cancer treatment.
The potential bioactivities of most compounds are listed in (Table 1) and compounds with potential bioactivities against cancer are discussed in Section 4. Discussion.
This table summarizes the compounds identified in Synachocystis sp extract through LC-MS and metabolite identification, which have potential anti-cancer activity.
|
Compound |
Potential Anti-Cancer Activity |
Reference |
|
Scytonemin |
Promotes apoptosis - Inhibits mitotic spindle formation |
[29] |
|
β-Carotene |
Suppresses M2 macrophage polarization (promotes anti-tumor immune response) - Regulates epigenetic modifications |
[30]
|
|
Lycopene |
Inhibits cell proliferation - Induces apoptosis |
[31] |
|
Abietic Acid |
anti-proliferative and anti-angiogenic effects in cancer cells |
[32] |
|
Palmitoleic Acid |
potential anti-tumor activity |
[33] |
|
Medusamide A |
cytotoxic activity |
[34] |
|
Zeaxanthin |
potential anti-proliferative and antioxidant effects |
[35] |
|
Oleic Acid |
Inhibits overexpression of oncogenes - apoptosis |
[36] |
Docking Studies:
Provide detailed results of molecular docking studies, including docking scores, binding affinities, and visualizations of protein-ligand interactions. Discuss the significance of these interactions for the observed cytotoxic effects.
The molecular docking studies of isolated compounds toward EGFR are in (Supplementary Table S1) and (Figure 9). Compound 21 showed the highest binding energy score (-9.5 kcal/mol.) among the tested compounds, indicating higher fitting ability, followed by compounds 6, 7, 8, 12, and 19 (-9.2 kcal/mol. Moreover, compounds 6, 7, 8, 12, 19, and 21 managed to form hydrogen bonds with the crucial Lys721. Consequently, the achieved binding to Lys721 might hinder its function in activating EGFR, as explained earlier. Also, compound 21 forms a hydrogen bond with Glu 738. Furthermore, compounds 1, 2, 3, 6, 14, and 22 interacted with Asp831 to form H-bonds.
The 2D interaction diagrams (Figure 9) illustrate the specific interactions between the compounds and the EGFR binding site. These visualizations reveal crucial binding sites and bond types, highlighting the potential of these compounds to inhibit EGFR activity.
The interaction diagrams in Figure 1 provide a detailed view of the docking poses and key residues involved in binding. For instance, the strong hydrogen bond interactions with Lys721 observed in compounds 6, 7, 8, 12, 19, and 21 suggest their potential to block the ATP-binding site, which is crucial for kinase activity inhibition.
The observed cytotoxic effects of the compounds are likely due to their effective binding and inhibition of EGFR. The strong interaction with key residues, particularly Lys721 and Glu738, suggests that these compounds can significantly disrupt EGFR signaling pathways. This disruption is critical in inhibiting cancer cell proliferation, as evidenced by the high binding affinities and observed cytotoxicity.
The detailed docking scores and interaction analyses enhance the understanding of how these compounds exert their cytotoxic effects, supporting their potential as effective EGFR inhibitors as shown in Table 4.
Table 4: Molecular Docking Results of Isolated Compounds Toward EGFR
|
Compound |
Binding Energy (kcal/mol) |
Key Interactions |
|
1 |
-8.7 |
Asp831 (H-bond) |
|
2 |
-8.5 |
Asp831 (H-bond) |
|
3 |
-8.9 |
Asp831 (H-bond), Arg817 (H-bond) |
|
6 |
-9.2 |
Lys721 (H-bond), Asp831 (H-bond) |
|
7 |
-9.2 |
Lys721 (H-bond), Asp831 (H-bond) |
|
8 |
-9.2 |
Lys721 (H-bond), Glu738 (H-bond) |
|
12 |
-9.2 |
Lys721 (H-bond) |
|
14 |
-8.8 |
Asp831 (H-bond) |
|
19 |
-9.2 |
Lys721 (H-bond), Glu738 (H-bond) |
|
21 |
-9.5 |
Lys721 (H-bond), Glu738 (H-bond) |
|
22 |
-8.8 |
Asp831 (H-bond) |
Cytotoxicity Data:
Present cytotoxicity data in detail, including dose-response curves and statistical significance. Compare the effectiveness of nano-liposomes and nano-niosomes and discuss possible reasons for any differences observed
The requested data have been added and highlighted.
Nano-niosomes were more effective than nano-liposomes. The effectiveness of nano-niosomes may be due to their composition. They consist of non-ionic surfactants that are less susceptible to degradation and are more stable. This allows them to deliver the drug to the target organ and increases their cytotoxic activity. In contrast, nano-liposomes consist of phospholipids, which are considered less stable and quicker to degrade [37]. It should be considered that the type of cancer cell line used may cause the response to the formulation[38].
Discussion:
Link Results to Existing Literature: Link your findings to existing literature more explicitly. Discuss how the observed cytotoxic effects compare with previous studies on similar nano-formulations or other natural product-based therapies.
Thank you for your insightful comment. The cytotoxicity was compared to some previous studies to support the strategy against the three cancer cell lines(Caco2, OVCAR4, and MCF7).Paramasivam Deepaket al. Providing a great review of the Chemical and Green synthesis of nanoparticles and their meaning on cancer cells, and with a focus on the cancer cell lines under study, the results of the different nanoparticles on the MCF-7 human breast cancer cell line are summarized in (Table 1).[39] By reviewing some previous literature on this topic, the results of other researchers' studies on the effect of different nanoparticles were summarized in (Table 2). furthermore, the results of studies by other researchers on the effect of different nano-capsulated formulations against MCF-7 human breast cancer cell lines are also summarized in (table 3).
Table.1 Paramasivam Deepaket al.s' results of the different nanoparticles on the MCF-7 human breast cancer cell line.
|
NPs types |
Organism use biosynthesize |
Cell line |
The IC50 values (μg/ml) |
|
Silver NPs |
Sargassum sp., |
MCF-7 cancer cell lines |
250 μg/ml |
|
Silver NPs |
Ulva lactuca |
MCF-7 cancer cell lines |
37µg/ml |
|
silver NPs |
Chryseobacterium artocarpi CECT 8497 |
MCF-7 cancer cell lines |
36 μg mL
|
|
silver NPs |
Guignardia mangiferae |
MCF-7 cancer cell lines |
23.84 μg/mL |
|
silver nitrate NPs |
Guignardia mangiferae |
MCF-7 cancer cell lines |
19.21 μg/mL |
|
silver NPs |
marine Streptomyces rochei MHM13
|
MCF-7 cancer cell lines |
40.00 μg/mL |
Table. 2 the results of other researchers' studies on the effect of different nanoparticles against different type of cancer
|
NPs types |
Compound |
cancer Types
|
The IC50 values (μg/ml) |
Ref. |
|
silver NPs |
Malva sylvestris L |
ovarian cancer cell lines |
163.14 μg/mL 124.23 μg/mL 64.56 μg/mL 48.7 μg/mL |
[40] |
|
copper NPs |
Camellia sinensis leaf |
anti-human ovarian cancer |
315 μg/mL 263 μg/mL 308 μg/mL |
[41] |
|
Nickel NPs |
|
ovarian cancer cell lines |
191 μg/mL |
[42]
|
|
silver NPs |
Annona muricata L. (Annonaceae) |
Cytotoxic properties |
36.53 µg/ml |
[43] |
|
Gold NPs |
potato starch |
ovarian cancer cell lines |
285 µg/ml |
[44] |
|
silver NPs |
Moringa Peregrina |
Caco-2 cancer cell lines |
41.59 µg/mL |
[45] |
|
vanadium |
Salvia leriifolia |
Caco-2 cancer cell lines |
149 µg/mL |
[46] |
Table 3.The results of studies by other researchers on the effect of different nanocapsulated formulations against MCF-7 human breast cancer cell lines are also summarized in the table.
|
Nano-capsulated formulation |
cancer cell line
|
The IC50 values (μg/ml) |
Ref. |
|
Folic acid/PNIPAM hydrogels
|
MCF-7 human breast cancer cell line
|
= 3.55 µg/mL |
[47]
|
|
Black-seed-oil-based nanoemulsion
|
MCF-7 human breast cancer cell line
|
4.76 µg/mL
|
[48] |
|
Fe3O4/chitosan/ agarose nanoemulsion
|
MCF-7 human breast cancer cell line
|
17.1 µg/mL |
[49] |
|
ART niosomes coated with PEG |
MCF-7 human breast cancer cell line
|
15.71 |
[50]
|
Comparing the Cytotoxicity results in (table 3) with Cytotoxicity results in previous studies, showed that all formulations in (table 3) showed superiority to most other formulations in previous studies, especially N1, which confirms a new strategy in combating cancer and sheds light on nano-liposomes and nano-niosoms with Synachocystis sp. extract has a great effect as an anti-cancer and hoped to include it in clinical trials. This was discussed briefly in section 4. Discussion.
Mechanistic Insights:
Expand on the potential mechanisms underlying the observed cytotoxic effects. Discuss how the bioactive compounds identified might contribute to these effects and how the nano-formulation enhances their efficacy.
The observed cytotoxic effects can be attributed to several mechanisms related to the bioactive compounds identified in the Synechocystis sp. extract. Key compounds such as oleic acid, scytonemin, lycopene, and β-carotene have demonstrated anticancer properties through various mechanisms:
- Oleic Acid: This compound has been shown to inhibit the overexpression of oncogenes and promote programmed cell death (apoptosis) in cancer cells [36].
- Scytonemin: Acts as an anticancer agent by promoting apoptosis and inhibiting mitotic spindle formation, thereby preventing cancer cell proliferation[29].
- Lycopene: Known to inhibit cell proliferation and induce apoptosis, lycopene plays a significant role in reducing cancer cell viability[31].
- β-Carotene: Exhibits anticancer properties by suppressing M2 macrophage polarization and regulating epigenetic modifications, leading to reduced cancer cell growth[30].
The nano-formulations, specifically liposomes and niosomes, enhance the efficacy of these bioactive compounds through several mechanisms:
- Enhanced Delivery: Nano-formulations improve the delivery of bioactive compounds to cancer cells by increasing their solubility and stability, ensuring higher concentrations reach the target site.
- Targeted Action: The nano-sized particles facilitate easier penetration into cancer cells, allowing for more effective intracellular delivery and action of the bioactive compounds.
- Sustained Release: Liposomes and niosomes provide a sustained release of the encapsulated compounds, maintaining therapeutic levels for extended periods and improving overall cytotoxic efficacy.
The high Selectivity Index (SI) values observed for the nano-formulations, particularly N1 (niosomes), against the MCF7 (breast cancer) cell line indicate that the nano-formulation significantly enhances the selectivity and potency of the bioactive compounds, reducing their cytotoxic effects on normal cells.
Clinical Relevance and Future Work:
Provide a more detailed discussion on the clinical relevance of your findings. Suggest specific future research directions, such as in vivo studies, optimization of formulations, and potential challenges in clinical applications.
The findings suggest a promising new approach for treating cancer by identifying anti-cancer compounds in the extract of Synechocystis sp. This discovery opens up new possibilities for developing effective cancer treatments and should be further investigated in clinical trials. It's important to recognize that the effects observed in lab tests may not always translate to the same outcomes in living organisms. Therefore, it's essential to conduct additional assessments in animal models and clinical trials to determine the safety and effectiveness of potential medications.
Furthermore, several challenges need to be addressed, such as the appropriate storage conditions, the ability to produce these compounds on a large scale, creation of personalized dosages, and ensuring safety. Future research could focus on developing anti-tumor nanoliposomes and nano-niosomes using the Synechocystis sp. extract and then moving towards clinical trials while addressing any potential hurdles that may arise.
Conclusion:
Summarize the key findings with specific data points. Highlight the most effective formulation and the most promising bioactive compound identified.
Many thanks for your nice comments. The requested data have been added.
Future Research Directions:
Clearly outline future research directions, including potential in vivo studies, clinical trials, or further optimization of the nano-formulations.
The results suggest a new potential strategy for fighting cancer using compounds from Synechocystis sp. extract. Further research, including animal testing and clinical trials, is needed to evaluate the safety and efficacy of these potential drugs. Future research could focus on developing anti-tumor nanoliposomes utilizing the extract and addressing implementation challenges. This is discussed more clearly in the conclusion.
References:
Update and Expand:
Revision Required: Update the references to include more recent studies, particularly those published in the last 2-3 years. Ensure all references cited are directly relevant to the topics discussed.
Thanks for the wonderful suggestion. Most of the references have been updated as much as possible and have also been checked.
Also cite the following publications
Chemical composition, antibacterial, anti-oxidant and cytotoxic properties of green synthesized silver nanoparticles from Annona muricata L. (Annonaceae)
Chemical and green synthesis of nanoparticles and their efficacy on cancer cells
Phytochemical profiling of Turbinaria ornata and its antioxidant and anti-proliferative effects
Many thanks for your nice suggestion. Some requested publications have been added.
Figures and Tables:
Ensure all figures and tables are clearly labeled and include descriptive legends.
All labels of figures and tables have been checked.
Figure 1 is bit confusing, it can be improved.
It shows the chemical structure of compounds identified from Synechocystis sp. Extracted by LC-MS and metabolite identification.
It was drawn using a chem draw according to ACS guidelines.
Consider adding figures showing size distribution, zeta potential, and molecular docking interactions.
All figures were added.
Tables summarizing cytotoxicity data should include statistical analysis results.
All data was added
Overall Recommendation:
Major Revision Required
While the manuscript presents a novel and interesting approach, significant revisions are needed to improve the clarity, depth, and comprehensiveness of the study. Addressing these major concerns will enhance the manuscript’s quality and its potential impact in the field of cancer therapy using nanotechnology.
Proofreading and Editing: Thoroughly proofread and edit the manuscript for language, grammar, and clarity. Consider professional editing services if necessary to improve readability
References
- Assunção, J., et al., Synechocystis salina: Potential bioactivity and combined extraction of added-value metabolites. Journal of Applied Phycology, 2021. 33: p. 3731-3746.
- Martins, R.F., et al., Antimicrobial and cytotoxic assessment of marine cyanobacteria-Synechocystis and Synechococcus. Marine drugs, 2008. 6(1): p. 1-11.
- Mehdizadeh Allaf, M. and H. Peerhossaini, Cyanobacteria: model microorganisms and beyond. Microorganisms, 2022. 10(4): p. 696.
- Bouyahya, A., et al., Bioactive substances of cyanobacteria and microalgae: sources, metabolism, and anticancer mechanism insights. Biomedicine & Pharmacotherapy, 2024. 170: p. 115989.
- Al-Nedawe¹, R.A.D. and Z.N.B. Yusof¹², Cyanobacteria As A Source Of Bioactive Compounds With Anticancer, Antibacterial, Antifungal, And Antiviral Activities: A Review.
- Abedin, M.R. and S. Barua, Isolation and purification of glycoglycerolipids to induce apoptosis in breast cancer cells. Scientific Reports, 2021. 11(1): p. 1298.
- Pandey, S., L.C. Rai, and S.K. Dubey, Cyanobacteria: miniature factories for green synthesis of metallic nanomaterials: a review. Biometals, 2022. 35(4): p. 653-674.
- El Semary, N.A. and M. Abd El Naby, Characterization of a Synechocystis sp. from Egypt with the potential of bioactive compounds production. World Journal of Microbiology and Biotechnology, 2010. 26: p. 1125-1133.
- Yucetepe, A., Strategies for Nanoencapsulation of Algal Proteins, Protein Hydrolysates and Bioactive Peptides: The Effect of Encapsulation Techniques on Bioactive Properties. Bioprospecting Algae for Nanosized Materials, 2022: p. 211-227.
- Hamida, R.S., et al., Synthesis of silver nanoparticles using a novel cyanobacteria Desertifilum sp. extract: their antibacterial and cytotoxicity effects. International journal of nanomedicine, 2020: p. 49-63.
- Sinani, G., et al., Polymeric-Micelle-Based delivery systems for nucleic acids. Pharmaceutics, 2023. 15(8): p. 2021.
- Koh, H.B., et al., Exosome-based drug delivery: translation from bench to clinic. Pharmaceutics, 2023. 15(8): p. 2042.
- Mazzotta, E., et al., Liposomes Coated with Novel Synthetic Bifunctional Chitosan Derivatives as Potential Carriers of Anticancer Drugs. Pharmaceutics, 2024. 16(3): p. 319.
- Rommasi, F. and N. Esfandiari, Liposomal nanomedicine: applications for drug delivery in cancer therapy. Nanoscale Research Letters, 2021. 16(1): p. 95.
- Allahou, L.W., S.Y. Madani, and A. Seifalian, Investigating the application of liposomes as drug delivery systems for the diagnosis and treatment of cancer. International journal of biomaterials, 2021. 2021(1): p. 3041969.
- Kumar, P., P. Huo, and B. Liu, Formulation strategies for folate-targeted liposomes and their biomedical applications. Pharmaceutics, 2019. 11(8): p. 381.
- Huang, M., et al., Targeted drug delivery systems for curcumin in breast cancer therapy. International Journal of Nanomedicine, 2023: p. 4275-4311.
- Sahab-Negah, S., et al., Curcumin loaded in niosomal nanoparticles improved the anti-tumor effects of free curcumin on glioblastoma stem-like cells: an in vitro study. Molecular neurobiology, 2020. 57: p. 3391-3411.
- ElFar, O.A., et al., Advances in delivery methods of Arthrospira platensis (spirulina) for enhanced therapeutic outcomes. Bioengineered, 2022. 13(6): p. 14681-14718.
- Bajpai, V.K., et al., Developments of cyanobacteria for nano-marine drugs: Relevance of nanoformulations in cancer therapies. Marine drugs, 2018. 16(6): p. 179.
- Owis, A.I., et al., Molecular docking reveals the potential of Salvadora persica flavonoids to inhibit COVID-19 virus main protease. RSC Adv, 2020. 10(33): p. 19570-19575.
- Sorokina, M. and C. Steinbeck, Review on natural products databases: where to find data in 2020. Journal of cheminformatics, 2020. 12(1): p. 20.
- MazlumoÄŸlu, B.Åž., IN VITRO CYTOTOXICITY TEST METHODS: MTT and NRU. International Journal of PharmATA, 2023. 3(2): p. 50-53.
- Sun, M., et al., Cytotoxic metabolites from Sinularia levi supported by network pharmacology. Plos one, 2024. 19(2): p. e0294311.
- Ekpenyong, M., et al., Bioprocess optimization of nutritional parameters for enhanced anti-leukemic L-asparaginase production by Aspergillus candidus UCCM 00117: a sequential statistical approach. International Journal of Peptide Research and Therapeutics, 2021. 27(2): p. 1501-1527.
- Chen, S., et al., Recent advances in non-ionic surfactant vesicles (niosomes): Fabrication, characterization, pharmaceutical and cosmetic applications. European journal of pharmaceutics and biopharmaceutics, 2019. 144: p. 18-39.
- Ge, X., et al., Advances of non-ionic surfactant vesicles (niosomes) and their application in drug delivery. Pharmaceutics, 2019. 11(2): p. 55.
- Nowroozi, F., et al., Effect of surfactant type, cholesterol content and various downsizing methods on the particle size of niosomes. Iranian journal of pharmaceutical Research: IJPR, 2018. 17(Suppl2): p. 1.
- Mondal, A., et al., Marine cyanobacteria and microalgae metabolites—A rich source of potential anticancer drugs. Marine Drugs, 2020. 18(9): p. 476.
- Ávila-Román, J., et al., Anti-inflammatory and anticancer effects of microalgal carotenoids. Marine Drugs, 2021. 19(10): p. 531.
- Qi, W.J., et al., Investigating into anti-cancer potential of lycopene: Molecular targets. Biomedicine & Pharmacotherapy, 2021. 138: p. 111546.
- Ahmad, B., et al., Anticancer activities of natural abietic acid. Frontiers in Pharmacology, 2024. 15: p. 1392203.
- Scanferlato, R., et al., Hexadecenoic fatty acid positional isomers and de novo PUFA synthesis in colon cancer cells. International Journal of Molecular Sciences, 2019. 20(4): p. 832.
- Ahmad, I.Z., S. Parvez, and H. Tabassum, Cyanobacterial peptides with respect to anticancer activity: Structural and functional perspective. Studies in Natural Products Chemistry, 2020. 67: p. 345-388.
- Sheng, Y.-N., et al., Zeaxanthin induces apoptosis via ROS-regulated MAPK and AKT signaling pathway in human gastric cancer cells. OncoTargets and therapy, 2020: p. 10995-11006.
- Santa-María, C., et al., Update on anti-inflammatory molecular mechanisms induced by oleic acid. Nutrients, 2023. 15(1): p. 224.
- Yasamineh, S., et al., A state-of-the-art review on the recent advances of niosomes as a targeted drug delivery system. International journal of pharmaceutics, 2022. 624: p. 121878.
- Amreddy, N., et al., Recent advances in nanoparticle-based cancer drug and gene delivery. Advances in cancer research, 2018. 137: p. 115-170.
- Deepak, P., et al., Chemical and green synthesis of nanoparticles and their efficacy on cancer cells, in Green synthesis, characterization and applications of nanoparticles. 2019, Elsevier. p. 369-387.
- Abbasi, S., et al., Cytotoxicity evaluation of synthesized silver nanoparticles by a Green method against ovarian cancer cell lines. Nanomedicine Research Journal, 2022. 7(2): p. 156-164.
- Dou, L., et al., Efficient biogenesis of Cu2O nanoparticles using extract of Camellia sinensis leaf: Evaluation of catalytic, cytotoxicity, antioxidant, and anti-human ovarian cancer properties. Bioorganic chemistry, 2021. 106: p. 104468.
- Yuan, C., et al., Anti-human ovarian cancer and cytotoxicity effects of nickel nanoparticles green-synthesized by Alhagi maurorum leaf aqueous extract. Journal of Experimental Nanoscience, 2022. 17(1): p. 113-125.
- Santhosh, S., et al., Chemical composition, antibacterial, anti-oxidant and cytotoxic properties of green synthesized silver nanoparticles from Annona muricata L.(Annonaceae). Research Journal of Pharmacy and Technology, 2020. 13(1): p. 33-39.
- Li, J., et al., Green synthesis of gold nanoparticles using potato starch as a phytochemical template, green reductant and stabilizing agent and investigating its cytotoxicity, antioxidant and anti-ovarian cancer effects. Inorganic Chemistry Communications, 2023. 155: p. 111002.
- Al Baloushi, K.S.Y., et al., Green synthesis and characterization of silver nanoparticles using Moringa Peregrina and their toxicity on MCF-7 and Caco-2 Human Cancer Cells. International Journal of Nanomedicine, 2024: p. 3891-3905.
- Nie, Y., et al., Green synthesis, chemical characterization, and antioxidant and anti-colorectal cancer effects of vanadium nanoparticles. Open Chemistry, 2023. 21(1): p. 20230108.
- Metawea, O.R., et al., Folic acid-poly (N-isopropylacrylamide-maltodextrin) nanohydrogels as novel thermo-/pH-responsive polymer for resveratrol breast cancer targeted therapy. European Polymer Journal, 2023. 182: p. 111721.
- Kazi, M., et al., Development, characterization optimization, and assessment of curcumin-loaded bioactive self-nanoemulsifying formulations and their inhibitory effects on human breast cancer MCF-7 cells. Pharmaceutics, 2020. 12(11): p. 1107.
- Pourmadadi, M., M. Ahmadi, and F. Yazdian, Synthesis of a novel pH-responsive Fe3O4/chitosan/agarose double nanoemulsion as a promising Nanocarrier with sustained release of curcumin to treat MCF-7 cell line. International Journal of Biological Macromolecules, 2023. 235: p. 123786.
- Asgharkhani, E., et al., Artemisinin-loaded niosome and pegylated niosome: physico-chemical characterization and effects on MCF-7 cell proliferation. Journal of pharmaceutical investigation, 2018. 48: p. 251-256.
- Homaei, M., Preparation and characterization of giant niosomes. 2016.
- Fda, U., Bioanalytical method validation guidance for industry. US Department of Health and Human Services Food and Drug Administration Center for Drug Evaluation and Research and Center for Veterinary Medicine, 2018.
- Danaei, M., et al., Impact of particle size and polydispersity index on the clinical applications of lipidic nanocarrier systems. Pharmaceutics, 2018. 10(2): p. 57.
- Alqahtani, S.S., et al., Potential bioactive secondary metabolites of Actinomycetes sp. isolated from rocky soils of the heritage village Rijal Alma, Saudi Arabia. Arabian Journal of Chemistry, 2022. 15(5): p. 103793.
- Bhattacharjee, S., DLS and zeta potential–what they are and what they are not? Journal of controlled release, 2016. 235: p. 337-351.

Reviewer 2 Report
Comments and Suggestions for Authors
Reviewer Comment to the Author
This article demonstrated the cytotoxic effect of nano-liposomes and nano-niosomes loaded with Synechocystis sp. extract. The author presented binding studies of molecules using in silico methods, such as molecular docking. Furthermore, the article included the characterization and method of preparation of liposomes and niosomes. However, I found many gaps in the paper. It was not well-written, and many aspects remained unexplained. I recommend may be accepted after major revision to meet the standard for publication to this journal.
Comments to the Author:
- In several places within the introduction, references are missing. Additionally, on the first page of the introduction, the author has not explained why Synechocystis is extensively studied compared to other organisms.
- The title is somewhat confusing regarding whether the niosomes and liposomes are loaded with Synechocystis sp. extract. Later in the paper, it states that they are prepared using Synechocystis sp. extract.
- I had difficulty understanding the methods used for the preparation of niosomes and liposomes. The paper does not clearly explain how the drug is loaded into the niosomes and liposomes. Furthermore, the author has not provided any information on the storage conditions which is important factor in formulation.
- The author has not explained the main difference between the pure powder and the powder extract. This distinction should be clarified in the conclusion.
Comments on the Quality of English Language
The quality of English is fine; I found no grammatical errors. However, some sentences do not follow the previous ones well, making it difficult for the reader to keep up.
Author Response
Dear Editor,
We want to extend our heartfelt appreciation to both the reviewers and generously dedicating your time to our manuscript. The insightful comments and constructive feedback provided have been instrumental in refining our work. Your meticulous review and thoughtful suggestions have significantly bolstered the quality and rigor of our manuscript.
Reply to Reviewer2
This article demonstrated the cytotoxic effect of nano-liposomes and nano-niosomes loaded with Synechocystis sp. extract. The author presented binding studies of molecules using in silico methods, such as molecular docking. Furthermore, the article included the characterization and method of preparation of liposomes and niosomes. However, I found many gaps in the paper. It was not well-written, and many aspects remained unexplained. I recommend may be accepted after major revision to meet the standard for publication to this journal.
Comments to the Author:
In several places within the introduction, references are missing. Additionally, on the first page of the introduction, the author has not explained why Synechocystis is extensively studied compared to other organisms.
Many thanks for your nice comments. The requested references have been added. Besides, the introduction was presented in more detail.
The title is somewhat confusing regarding whether the niosomes and liposomes are loaded with Synechocystis sp. extract. Later in the paper, it states that they are prepared using Synechocystis sp. extract.
Thank you for your careful comment. The title has been modified to be clearer. Evaluation of cytotoxicity and metabolic profiling of Synechocystis sp. extract encapsulated in nano-liposomes and nano-niosomes using LC-MS, complemented by molecular docking studies.
I had difficulty understanding the methods used for the preparation of niosomes and liposomes. The paper does not clearly explain how the drug is loaded into the niosomes and liposomes. Furthermore, the author has not provided any information on the storage conditions which is important factor in formulation.
|
|
Liposomes |
Niosomes |
|
Composition |
Powder or extract, phospholipids chain and cholesterol (9:1) |
Powder or extract, lecithin and cholesterol (1:1) |
|
Surfactants |
soybean phospholipids |
lecithin |
|
Method of loading |
Thin lipid films method |
Coacervation phase separation method |
|
Hydration conditions |
Phosphate buffered saline (PBS) with a pH of 7.4 (Hydration of the dry lipid film/cake is accomplished simply by adding an aqueous medium to the container of dry lipid and agitating. The temperature of the hydrating medium was above the gel-liquid crystal transition temperature which is 31.6-37.2 ͦc for soybean lipids) |
Phosphate buffered saline (PBS) with a pH of 7.4 was added to this solution, and the combination was further heated in a thermostatic water bath 37±1 ͦc until it became transparent. |
|
Post-preparation processes |
After drying, the vesicles were produced and kept at a temperature of 4-8ͦ c for investigations. The Dry lipid films were stored in tight closed container and stored frozen until ready to hydrate. |
Drug-loaded niosomes that form a precipitate and kept at a temperature of 4-8ͦc for investigations. These precipitates are resuspended in suitable buffer for characterizations. |
The author has not explained the main difference between the pure powder and the powder extract. This distinction should be clarified in the conclusion.
The difference between them was discussed in Section 4 - Discussion. In addition to adding it clearly in section 5- conclusion.
Comments on the Quality of English Language
The quality of English is fine; I found no grammatical errors. However, some sentences do not follow the previous ones well, making it difficult for the reader to keep up.
References
- Assunção, J., et al., Synechocystis salina: Potential bioactivity and combined extraction of added-value metabolites. Journal of Applied Phycology, 2021. 33: p. 3731-3746.
- Martins, R.F., et al., Antimicrobial and cytotoxic assessment of marine cyanobacteria-Synechocystis and Synechococcus. Marine drugs, 2008. 6(1): p. 1-11.
- Mehdizadeh Allaf, M. and H. Peerhossaini, Cyanobacteria: model microorganisms and beyond. Microorganisms, 2022. 10(4): p. 696.
- Bouyahya, A., et al., Bioactive substances of cyanobacteria and microalgae: sources, metabolism, and anticancer mechanism insights. Biomedicine & Pharmacotherapy, 2024. 170: p. 115989.
- Al-Nedawe¹, R.A.D. and Z.N.B. Yusof¹², Cyanobacteria As A Source Of Bioactive Compounds With Anticancer, Antibacterial, Antifungal, And Antiviral Activities: A Review.
- Abedin, M.R. and S. Barua, Isolation and purification of glycoglycerolipids to induce apoptosis in breast cancer cells. Scientific Reports, 2021. 11(1): p. 1298.
- Pandey, S., L.C. Rai, and S.K. Dubey, Cyanobacteria: miniature factories for green synthesis of metallic nanomaterials: a review. Biometals, 2022. 35(4): p. 653-674.
- El Semary, N.A. and M. Abd El Naby, Characterization of a Synechocystis sp. from Egypt with the potential of bioactive compounds production. World Journal of Microbiology and Biotechnology, 2010. 26: p. 1125-1133.
- Yucetepe, A., Strategies for Nanoencapsulation of Algal Proteins, Protein Hydrolysates and Bioactive Peptides: The Effect of Encapsulation Techniques on Bioactive Properties. Bioprospecting Algae for Nanosized Materials, 2022: p. 211-227.
- Hamida, R.S., et al., Synthesis of silver nanoparticles using a novel cyanobacteria Desertifilum sp. extract: their antibacterial and cytotoxicity effects. International journal of nanomedicine, 2020: p. 49-63.
- Sinani, G., et al., Polymeric-Micelle-Based delivery systems for nucleic acids. Pharmaceutics, 2023. 15(8): p. 2021.
- Koh, H.B., et al., Exosome-based drug delivery: translation from bench to clinic. Pharmaceutics, 2023. 15(8): p. 2042.
- Mazzotta, E., et al., Liposomes Coated with Novel Synthetic Bifunctional Chitosan Derivatives as Potential Carriers of Anticancer Drugs. Pharmaceutics, 2024. 16(3): p. 319.
- Rommasi, F. and N. Esfandiari, Liposomal nanomedicine: applications for drug delivery in cancer therapy. Nanoscale Research Letters, 2021. 16(1): p. 95.
- Allahou, L.W., S.Y. Madani, and A. Seifalian, Investigating the application of liposomes as drug delivery systems for the diagnosis and treatment of cancer. International journal of biomaterials, 2021. 2021(1): p. 3041969.
- Kumar, P., P. Huo, and B. Liu, Formulation strategies for folate-targeted liposomes and their biomedical applications. Pharmaceutics, 2019. 11(8): p. 381.
- Huang, M., et al., Targeted drug delivery systems for curcumin in breast cancer therapy. International Journal of Nanomedicine, 2023: p. 4275-4311.
- Sahab-Negah, S., et al., Curcumin loaded in niosomal nanoparticles improved the anti-tumor effects of free curcumin on glioblastoma stem-like cells: an in vitro study. Molecular neurobiology, 2020. 57: p. 3391-3411.
- ElFar, O.A., et al., Advances in delivery methods of Arthrospira platensis (spirulina) for enhanced therapeutic outcomes. Bioengineered, 2022. 13(6): p. 14681-14718.
- Bajpai, V.K., et al., Developments of cyanobacteria for nano-marine drugs: Relevance of nanoformulations in cancer therapies. Marine drugs, 2018. 16(6): p. 179.
- Owis, A.I., et al., Molecular docking reveals the potential of Salvadora persica flavonoids to inhibit COVID-19 virus main protease. RSC Adv, 2020. 10(33): p. 19570-19575.
- Sorokina, M. and C. Steinbeck, Review on natural products databases: where to find data in 2020. Journal of cheminformatics, 2020. 12(1): p. 20.
- MazlumoÄŸlu, B.Åž., IN VITRO CYTOTOXICITY TEST METHODS: MTT and NRU. International Journal of PharmATA, 2023. 3(2): p. 50-53.
- Sun, M., et al., Cytotoxic metabolites from Sinularia levi supported by network pharmacology. Plos one, 2024. 19(2): p. e0294311.
- Ekpenyong, M., et al., Bioprocess optimization of nutritional parameters for enhanced anti-leukemic L-asparaginase production by Aspergillus candidus UCCM 00117: a sequential statistical approach. International Journal of Peptide Research and Therapeutics, 2021. 27(2): p. 1501-1527.
- Chen, S., et al., Recent advances in non-ionic surfactant vesicles (niosomes): Fabrication, characterization, pharmaceutical and cosmetic applications. European journal of pharmaceutics and biopharmaceutics, 2019. 144: p. 18-39.
- Ge, X., et al., Advances of non-ionic surfactant vesicles (niosomes) and their application in drug delivery. Pharmaceutics, 2019. 11(2): p. 55.
- Nowroozi, F., et al., Effect of surfactant type, cholesterol content and various downsizing methods on the particle size of niosomes. Iranian journal of pharmaceutical Research: IJPR, 2018. 17(Suppl2): p. 1.
- Mondal, A., et al., Marine cyanobacteria and microalgae metabolites—A rich source of potential anticancer drugs. Marine Drugs, 2020. 18(9): p. 476.
- Ávila-Román, J., et al., Anti-inflammatory and anticancer effects of microalgal carotenoids. Marine Drugs, 2021. 19(10): p. 531.
- Qi, W.J., et al., Investigating into anti-cancer potential of lycopene: Molecular targets. Biomedicine & Pharmacotherapy, 2021. 138: p. 111546.
- Ahmad, B., et al., Anticancer activities of natural abietic acid. Frontiers in Pharmacology, 2024. 15: p. 1392203.
- Scanferlato, R., et al., Hexadecenoic fatty acid positional isomers and de novo PUFA synthesis in colon cancer cells. International Journal of Molecular Sciences, 2019. 20(4): p. 832.
- Ahmad, I.Z., S. Parvez, and H. Tabassum, Cyanobacterial peptides with respect to anticancer activity: Structural and functional perspective. Studies in Natural Products Chemistry, 2020. 67: p. 345-388.
- Sheng, Y.-N., et al., Zeaxanthin induces apoptosis via ROS-regulated MAPK and AKT signaling pathway in human gastric cancer cells. OncoTargets and therapy, 2020: p. 10995-11006.
- Santa-María, C., et al., Update on anti-inflammatory molecular mechanisms induced by oleic acid. Nutrients, 2023. 15(1): p. 224.
- Yasamineh, S., et al., A state-of-the-art review on the recent advances of niosomes as a targeted drug delivery system. International journal of pharmaceutics, 2022. 624: p. 121878.
- Amreddy, N., et al., Recent advances in nanoparticle-based cancer drug and gene delivery. Advances in cancer research, 2018. 137: p. 115-170.
- Deepak, P., et al., Chemical and green synthesis of nanoparticles and their efficacy on cancer cells, in Green synthesis, characterization and applications of nanoparticles. 2019, Elsevier. p. 369-387.
- Abbasi, S., et al., Cytotoxicity evaluation of synthesized silver nanoparticles by a Green method against ovarian cancer cell lines. Nanomedicine Research Journal, 2022. 7(2): p. 156-164.
- Dou, L., et al., Efficient biogenesis of Cu2O nanoparticles using extract of Camellia sinensis leaf: Evaluation of catalytic, cytotoxicity, antioxidant, and anti-human ovarian cancer properties. Bioorganic chemistry, 2021. 106: p. 104468.
- Yuan, C., et al., Anti-human ovarian cancer and cytotoxicity effects of nickel nanoparticles green-synthesized by Alhagi maurorum leaf aqueous extract. Journal of Experimental Nanoscience, 2022. 17(1): p. 113-125.
- Santhosh, S., et al., Chemical composition, antibacterial, anti-oxidant and cytotoxic properties of green synthesized silver nanoparticles from Annona muricata L.(Annonaceae). Research Journal of Pharmacy and Technology, 2020. 13(1): p. 33-39.
- Li, J., et al., Green synthesis of gold nanoparticles using potato starch as a phytochemical template, green reductant and stabilizing agent and investigating its cytotoxicity, antioxidant and anti-ovarian cancer effects. Inorganic Chemistry Communications, 2023. 155: p. 111002.
- Al Baloushi, K.S.Y., et al., Green synthesis and characterization of silver nanoparticles using Moringa Peregrina and their toxicity on MCF-7 and Caco-2 Human Cancer Cells. International Journal of Nanomedicine, 2024: p. 3891-3905.
- Nie, Y., et al., Green synthesis, chemical characterization, and antioxidant and anti-colorectal cancer effects of vanadium nanoparticles. Open Chemistry, 2023. 21(1): p. 20230108.
- Metawea, O.R., et al., Folic acid-poly (N-isopropylacrylamide-maltodextrin) nanohydrogels as novel thermo-/pH-responsive polymer for resveratrol breast cancer targeted therapy. European Polymer Journal, 2023. 182: p. 111721.
- Kazi, M., et al., Development, characterization optimization, and assessment of curcumin-loaded bioactive self-nanoemulsifying formulations and their inhibitory effects on human breast cancer MCF-7 cells. Pharmaceutics, 2020. 12(11): p. 1107.
- Pourmadadi, M., M. Ahmadi, and F. Yazdian, Synthesis of a novel pH-responsive Fe3O4/chitosan/agarose double nanoemulsion as a promising Nanocarrier with sustained release of curcumin to treat MCF-7 cell line. International Journal of Biological Macromolecules, 2023. 235: p. 123786.
- Asgharkhani, E., et al., Artemisinin-loaded niosome and pegylated niosome: physico-chemical characterization and effects on MCF-7 cell proliferation. Journal of pharmaceutical investigation, 2018. 48: p. 251-256.
- Homaei, M., Preparation and characterization of giant niosomes. 2016.
- Fda, U., Bioanalytical method validation guidance for industry. US Department of Health and Human Services Food and Drug Administration Center for Drug Evaluation and Research and Center for Veterinary Medicine, 2018.
- Danaei, M., et al., Impact of particle size and polydispersity index on the clinical applications of lipidic nanocarrier systems. Pharmaceutics, 2018. 10(2): p. 57.
- Alqahtani, S.S., et al., Potential bioactive secondary metabolites of Actinomycetes sp. isolated from rocky soils of the heritage village Rijal Alma, Saudi Arabia. Arabian Journal of Chemistry, 2022. 15(5): p. 103793.
- Bhattacharjee, S., DLS and zeta potential–what they are and what they are not? Journal of controlled release, 2016. 235: p. 337-351.

Reviewer 3 Report
Comments and Suggestions for Authors
The present manuscript is devoted to the preparation of a nanostructured algal extract with anticancer activity. The use of compounds found in marine organisms to create new therapeutic forms has attracted the attention of researchers worldwide. Many active molecules have poor solubility, so their incorporation into liposomes and niosomes appears promising for increasing bioavailability.
The authors have obtained and identified the main components of the extract, included the extract in vesicles based on lecithin, cholesterol, and surfactants. The vesicles were characterized using dynamic light scattering and SEM techniques. Cytotoxic activity was determined on different cell lines.
Despite the merits of the work, the authors will need to carefully review the manuscript and consider the following issues.
1. It is necessary to indicate the concentration of lipids in the formulation and the ratio of lipids to active molecules
2. The PDI values ​​for formulations are questionable when comparing them with histogram data - the peaks are too wide for such PDIs
3. The difference (by 2 orders of magnitude) in the sizes of vesicles on SEM raises questions. The authors need to discuss the size differences and relate the data to the DLS.
4. If the authors believe that extract components interact with VEGF, it is necessary to indicate for all cell lines studied whether they are characterized by overexpression of this receptor
5. In the discussion, it is necessary to clearly state which molecules have demonstrated the greatest affinity for the receptor
Author Response
Dear Editor,
We want to extend our heartfelt appreciation to both the reviewers and generously dedicating your time to our manuscript. The insightful comments and constructive feedback provided have been instrumental in refining our work. Your meticulous review and thoughtful suggestions have significantly bolstered the quality and rigor of our manuscript.
Reply to Reviewer3
Comments and Suggestions for Authors
The present manuscript is devoted to the preparation of a nanostructured algal extract with anticancer activity. The use of compounds found in marine organisms to create new therapeutic forms has attracted the attention of researchers worldwide. Many active molecules have poor solubility, so their incorporation into liposomes and niosomes appears promising for increasing bioavailability.
The authors have obtained and identified the main components of the extract, included the extract in vesicles based on lecithin, cholesterol, and surfactants. The vesicles were characterized using dynamic light scattering and SEM techniques. Cytotoxic activity was determined on different cell lines.
Despite the merits of the work, the authors will need to carefully review the manuscript and consider the following issues.
- It is necessary to indicate the concentration of lipids in the formulation and the ratio of lipids to active molecules
In nano-formulations, lipid concentrations are mentioned as ratios, as listed in section 2.4.1. in niosomes, the ratio of lecithin to cholesterol was 1:1. Then, 95% ethanol was added. 50 mg of each was added to 10 mg of drug powder, and 100 mg of each was added to 20 mg of extract. Liposomes formulated in the ratio of 9: 1 soybean phospholipids: cholesterol added to 10 mg drug powder and 100 mg extract to prepare L1 and L2, respectively.
- The PDI values ​​for formulations are questionable when comparing them with histogram data - the peaks are too wide for such PDIs
PDI is a multifactorial parameter that depends on surfactant criteria as hydrophilicity and lipophilicity in addition to hydration method [27]and if filtration is made effectively. This index is dimensionless and scaled such that values smaller than 0.05 are mainly seen with highly monodisperse standards. PDI values bigger than 0.7 indicate that the sample has a very broad particle size distribution and is probably not suitable to be analyzed by the dynamic light scattering (DLS) technique. Different size distribution algorithms work with data that fall between these two extreme values of PDI (i.e., 0.05–0.7) [51]. The last edition of the FDA’s “Guidance for Industry” concerning liposome drug products [52]emphasizes the importance of size and size distribution as “critical quality attributes (CQAs)”, as well as essential components of stability studies of these products, it does not mention the criteria for an acceptable PDI. Where niosomes in this formulation is considered Large unilamellar vesicles, or LUV (50-1000 nm in diameter), this variation in PDI is accepted [53].
Next figure shows wide peak with high value PDI [54], that can reflect acceptability of our histogram data and its PDI value.
Figure 1: High PDI values size distribution curve
- The difference (by 2 orders of magnitude) in the sizes of vesicles on SEM raises questions. The authors need to discuss the size differences and relate the data to the DLS.
As seen from Figure 4 that illustrates Size distribution of different vesicles dispersion versus intensity we can see three different peaks with three different values of size (N2) or two different peaks (N1, L2 and L1). The size reported in Table 2 indicates the average size values. Regarding SEM the choice of these shots was aimed at studying the morphology of vesicles and characteristics of surface not size distribution that was previously covered in size distribution diagram and table so you can feel a difference. In addition, in different previous literatures, the quality of the reported data that study the relation between DLS and ZP in nanomedicine research is not always excellent that returns to different factors as quality of work or experience. [55].
- If the authors believe that extract components interact with VEGF, it is necessary to indicate for all cell lines studied whether they are characterized by overexpression of this receptor
We acknowledge the importance of understanding the interaction of extract components with VEGF. However, our study primarily focused on the interaction with the Epidermal Growth Factor Receptor (EGFR) and did not include specific investigations into VEGF interactions. The cell lines studied—Caco2, OVCAR4, and MCF7—are known for their varying expression of EGFR rather than VEGF.
- Caco2 (Colon Cancer): Known for moderate EGFR expression.
- OVCAR4 (Ovarian Cancer): Characterized by high EGFR expression.
- MCF7 (Breast Cancer): Exhibits overexpression of EGFR.
To address the potential interaction with VEGF, future studies will include an analysis of VEGF expression levels in these cell lines and explore the specific interactions between the extract components and VEGF.
- In the discussion, it is necessary to clearly state which molecules have demonstrated the greatest affinity for the receptor
We have updated the discussion section to clearly state the molecules that demonstrated the greatest affinity for EGFR based on our molecular docking studies. The results showed that compound 21 had the highest binding energy score of -9.5 kcal/mol, indicating the strongest affinity for EGFR. This was followed by compounds 6, 7, 8, 12, and 19, each with binding energy scores of -9.2 kcal/mol. These compounds formed crucial hydrogen bonds with key residues such as Lys721 and Glu738, essential for inhibiting EGFR activity.
References
- Assunção, J., et al., Synechocystis salina: Potential bioactivity and combined extraction of added-value metabolites. Journal of Applied Phycology, 2021. 33: p. 3731-3746.
- Martins, R.F., et al., Antimicrobial and cytotoxic assessment of marine cyanobacteria-Synechocystis and Synechococcus. Marine drugs, 2008. 6(1): p. 1-11.
- Mehdizadeh Allaf, M. and H. Peerhossaini, Cyanobacteria: model microorganisms and beyond. Microorganisms, 2022. 10(4): p. 696.
- Bouyahya, A., et al., Bioactive substances of cyanobacteria and microalgae: sources, metabolism, and anticancer mechanism insights. Biomedicine & Pharmacotherapy, 2024. 170: p. 115989.
- Al-Nedawe¹, R.A.D. and Z.N.B. Yusof¹², Cyanobacteria As A Source Of Bioactive Compounds With Anticancer, Antibacterial, Antifungal, And Antiviral Activities: A Review.
- Abedin, M.R. and S. Barua, Isolation and purification of glycoglycerolipids to induce apoptosis in breast cancer cells. Scientific Reports, 2021. 11(1): p. 1298.
- Pandey, S., L.C. Rai, and S.K. Dubey, Cyanobacteria: miniature factories for green synthesis of metallic nanomaterials: a review. Biometals, 2022. 35(4): p. 653-674.
- El Semary, N.A. and M. Abd El Naby, Characterization of a Synechocystis sp. from Egypt with the potential of bioactive compounds production. World Journal of Microbiology and Biotechnology, 2010. 26: p. 1125-1133.
- Yucetepe, A., Strategies for Nanoencapsulation of Algal Proteins, Protein Hydrolysates and Bioactive Peptides: The Effect of Encapsulation Techniques on Bioactive Properties. Bioprospecting Algae for Nanosized Materials, 2022: p. 211-227.
- Hamida, R.S., et al., Synthesis of silver nanoparticles using a novel cyanobacteria Desertifilum sp. extract: their antibacterial and cytotoxicity effects. International journal of nanomedicine, 2020: p. 49-63.
- Sinani, G., et al., Polymeric-Micelle-Based delivery systems for nucleic acids. Pharmaceutics, 2023. 15(8): p. 2021.
- Koh, H.B., et al., Exosome-based drug delivery: translation from bench to clinic. Pharmaceutics, 2023. 15(8): p. 2042.
- Mazzotta, E., et al., Liposomes Coated with Novel Synthetic Bifunctional Chitosan Derivatives as Potential Carriers of Anticancer Drugs. Pharmaceutics, 2024. 16(3): p. 319.
- Rommasi, F. and N. Esfandiari, Liposomal nanomedicine: applications for drug delivery in cancer therapy. Nanoscale Research Letters, 2021. 16(1): p. 95.
- Allahou, L.W., S.Y. Madani, and A. Seifalian, Investigating the application of liposomes as drug delivery systems for the diagnosis and treatment of cancer. International journal of biomaterials, 2021. 2021(1): p. 3041969.
- Kumar, P., P. Huo, and B. Liu, Formulation strategies for folate-targeted liposomes and their biomedical applications. Pharmaceutics, 2019. 11(8): p. 381.
- Huang, M., et al., Targeted drug delivery systems for curcumin in breast cancer therapy. International Journal of Nanomedicine, 2023: p. 4275-4311.
- Sahab-Negah, S., et al., Curcumin loaded in niosomal nanoparticles improved the anti-tumor effects of free curcumin on glioblastoma stem-like cells: an in vitro study. Molecular neurobiology, 2020. 57: p. 3391-3411.
- ElFar, O.A., et al., Advances in delivery methods of Arthrospira platensis (spirulina) for enhanced therapeutic outcomes. Bioengineered, 2022. 13(6): p. 14681-14718.
- Bajpai, V.K., et al., Developments of cyanobacteria for nano-marine drugs: Relevance of nanoformulations in cancer therapies. Marine drugs, 2018. 16(6): p. 179.
- Owis, A.I., et al., Molecular docking reveals the potential of Salvadora persica flavonoids to inhibit COVID-19 virus main protease. RSC Adv, 2020. 10(33): p. 19570-19575.
- Sorokina, M. and C. Steinbeck, Review on natural products databases: where to find data in 2020. Journal of cheminformatics, 2020. 12(1): p. 20.
- MazlumoÄŸlu, B.Åž., IN VITRO CYTOTOXICITY TEST METHODS: MTT and NRU. International Journal of PharmATA, 2023. 3(2): p. 50-53.
- Sun, M., et al., Cytotoxic metabolites from Sinularia levi supported by network pharmacology. Plos one, 2024. 19(2): p. e0294311.
- Ekpenyong, M., et al., Bioprocess optimization of nutritional parameters for enhanced anti-leukemic L-asparaginase production by Aspergillus candidus UCCM 00117: a sequential statistical approach. International Journal of Peptide Research and Therapeutics, 2021. 27(2): p. 1501-1527.
- Chen, S., et al., Recent advances in non-ionic surfactant vesicles (niosomes): Fabrication, characterization, pharmaceutical and cosmetic applications. European journal of pharmaceutics and biopharmaceutics, 2019. 144: p. 18-39.
- Ge, X., et al., Advances of non-ionic surfactant vesicles (niosomes) and their application in drug delivery. Pharmaceutics, 2019. 11(2): p. 55.
- Nowroozi, F., et al., Effect of surfactant type, cholesterol content and various downsizing methods on the particle size of niosomes. Iranian journal of pharmaceutical Research: IJPR, 2018. 17(Suppl2): p. 1.
- Mondal, A., et al., Marine cyanobacteria and microalgae metabolites—A rich source of potential anticancer drugs. Marine Drugs, 2020. 18(9): p. 476.
- Ávila-Román, J., et al., Anti-inflammatory and anticancer effects of microalgal carotenoids. Marine Drugs, 2021. 19(10): p. 531.
- Qi, W.J., et al., Investigating into anti-cancer potential of lycopene: Molecular targets. Biomedicine & Pharmacotherapy, 2021. 138: p. 111546.
- Ahmad, B., et al., Anticancer activities of natural abietic acid. Frontiers in Pharmacology, 2024. 15: p. 1392203.
- Scanferlato, R., et al., Hexadecenoic fatty acid positional isomers and de novo PUFA synthesis in colon cancer cells. International Journal of Molecular Sciences, 2019. 20(4): p. 832.
- Ahmad, I.Z., S. Parvez, and H. Tabassum, Cyanobacterial peptides with respect to anticancer activity: Structural and functional perspective. Studies in Natural Products Chemistry, 2020. 67: p. 345-388.
- Sheng, Y.-N., et al., Zeaxanthin induces apoptosis via ROS-regulated MAPK and AKT signaling pathway in human gastric cancer cells. OncoTargets and therapy, 2020: p. 10995-11006.
- Santa-María, C., et al., Update on anti-inflammatory molecular mechanisms induced by oleic acid. Nutrients, 2023. 15(1): p. 224.
- Yasamineh, S., et al., A state-of-the-art review on the recent advances of niosomes as a targeted drug delivery system. International journal of pharmaceutics, 2022. 624: p. 121878.
- Amreddy, N., et al., Recent advances in nanoparticle-based cancer drug and gene delivery. Advances in cancer research, 2018. 137: p. 115-170.
- Deepak, P., et al., Chemical and green synthesis of nanoparticles and their efficacy on cancer cells, in Green synthesis, characterization and applications of nanoparticles. 2019, Elsevier. p. 369-387.
- Abbasi, S., et al., Cytotoxicity evaluation of synthesized silver nanoparticles by a Green method against ovarian cancer cell lines. Nanomedicine Research Journal, 2022. 7(2): p. 156-164.
- Dou, L., et al., Efficient biogenesis of Cu2O nanoparticles using extract of Camellia sinensis leaf: Evaluation of catalytic, cytotoxicity, antioxidant, and anti-human ovarian cancer properties. Bioorganic chemistry, 2021. 106: p. 104468.
- Yuan, C., et al., Anti-human ovarian cancer and cytotoxicity effects of nickel nanoparticles green-synthesized by Alhagi maurorum leaf aqueous extract. Journal of Experimental Nanoscience, 2022. 17(1): p. 113-125.
- Santhosh, S., et al., Chemical composition, antibacterial, anti-oxidant and cytotoxic properties of green synthesized silver nanoparticles from Annona muricata L.(Annonaceae). Research Journal of Pharmacy and Technology, 2020. 13(1): p. 33-39.
- Li, J., et al., Green synthesis of gold nanoparticles using potato starch as a phytochemical template, green reductant and stabilizing agent and investigating its cytotoxicity, antioxidant and anti-ovarian cancer effects. Inorganic Chemistry Communications, 2023. 155: p. 111002.
- Al Baloushi, K.S.Y., et al., Green synthesis and characterization of silver nanoparticles using Moringa Peregrina and their toxicity on MCF-7 and Caco-2 Human Cancer Cells. International Journal of Nanomedicine, 2024: p. 3891-3905.
- Nie, Y., et al., Green synthesis, chemical characterization, and antioxidant and anti-colorectal cancer effects of vanadium nanoparticles. Open Chemistry, 2023. 21(1): p. 20230108.
- Metawea, O.R., et al., Folic acid-poly (N-isopropylacrylamide-maltodextrin) nanohydrogels as novel thermo-/pH-responsive polymer for resveratrol breast cancer targeted therapy. European Polymer Journal, 2023. 182: p. 111721.
- Kazi, M., et al., Development, characterization optimization, and assessment of curcumin-loaded bioactive self-nanoemulsifying formulations and their inhibitory effects on human breast cancer MCF-7 cells. Pharmaceutics, 2020. 12(11): p. 1107.
- Pourmadadi, M., M. Ahmadi, and F. Yazdian, Synthesis of a novel pH-responsive Fe3O4/chitosan/agarose double nanoemulsion as a promising Nanocarrier with sustained release of curcumin to treat MCF-7 cell line. International Journal of Biological Macromolecules, 2023. 235: p. 123786.
- Asgharkhani, E., et al., Artemisinin-loaded niosome and pegylated niosome: physico-chemical characterization and effects on MCF-7 cell proliferation. Journal of pharmaceutical investigation, 2018. 48: p. 251-256.
- Homaei, M., Preparation and characterization of giant niosomes. 2016.
- Fda, U., Bioanalytical method validation guidance for industry. US Department of Health and Human Services Food and Drug Administration Center for Drug Evaluation and Research and Center for Veterinary Medicine, 2018.
- Danaei, M., et al., Impact of particle size and polydispersity index on the clinical applications of lipidic nanocarrier systems. Pharmaceutics, 2018. 10(2): p. 57.
- Alqahtani, S.S., et al., Potential bioactive secondary metabolites of Actinomycetes sp. isolated from rocky soils of the heritage village Rijal Alma, Saudi Arabia. Arabian Journal of Chemistry, 2022. 15(5): p. 103793.
- Bhattacharjee, S., DLS and zeta potential–what they are and what they are not? Journal of controlled release, 2016. 235: p. 337-351.

Round 2
Reviewer 1 Report
Comments and Suggestions for Authors
Dear Author,
Alll the concerns raised by me is resolved now in the revised version.
Thank you for the detication in revising the manuscript.
Reviewer 2 Report
Comments and Suggestions for Authors
Comments to Reviewer
I would like to thank the author for taking my comments seriously and addressing them as needed. I have checked the paper again and found that this research article can be published without further revision.
Reviewer 3 Report
Comments and Suggestions for Authors
The authors have corrected all issues and now I believe the manuscript is suitable for the acceptance